



# Effects of vertical grid spacing on the climate simulated in the ICON-Sapphire global storm-resolving model

Hauke Schmidt[1], Sebastian Rast[1], Jiawei Bao[1], Shih-Wei Fang[1], Diego Jimenez-de la Cuesta[1,2],
Paul Keil[1,3], Lukas Kluft[1], Clarissa Kroll[1,4], Theresa Lang[1], Ulrike Niemeier[1], Andrea Schneidereit[1,2],
Andrew I. L. Williams[5], and Bjorn Stevens[1]

[1]Max Planck Institute for Meteorology, Bundesstr. 53, 20146 Hamburg, Germany
[2]now at: Deutscher Wetterdienst (DWD), Offenbach, Germany
[3]now at: German Climate Computing Centre (DKRZ), Hamburg, Germany
[4]now at: Institute for Atmospheric and Climate Science, ETH Zürich, Zürich, Switzerland
[5]Atmospheric, Oceanic and Planetary Physics, Department of Physics, University of Oxford, Oxford, UK

**Correspondence:** Hauke Schmidt (hauke.schmidt@mpimet.mpg.de)

**Abstract.** Global storm-resolving models (GSRM) use strongly refined horizontal grids in comparison to climate models typically used in the Coupled Model Intercomparison Project (CMIP) but comparable vertical grid spacings. Here, we study how changes in vertical grid spacing and adjustments of the integration time step affect basic climate quantities simulated by the ICON-Sapphire atmospheric GSRM. Simulations are performed over a 45-day period for five different vertical grids having

between 55 and 540 vertical layers and maximum tropospheric vertical grid spacings between 800 and $50\,\mathrm{m}$. The effects of changes in vertical grid spacing are compared to differences between simulations with horizontal grid spacings of 5 and $2.5\,\mathrm{km}$. For most quantities considered, halving vertical grid spacing has smaller effects than halving horizontal grid spacing but is not negligible. Every halving of the vertical grid spacing jointly with the necessary reductions of the time step length increases cloud liquid water by about 7%, compared to about 16% decrease for halving the horizontal grid spacing. The effect is due

to both vertical grid refinement and time step reduction. There is no tendency of convergence in the range of grid spacings tested here. The cloud ice amount also increases with a refinement of the vertical grid but is hardly affected by the time step length and does show a tendency of convergence. While the effect on shortwave radiation is globally dominated by the changed reflection due to the changed cloud liquid water content, effects on longwave radiation are more difficult to interpret because changes in cloud ice concentration and cloud fraction are anticorrelated in some regions.

## 1 Introduction

A recent development in numerical simulations of the global atmosphere is the increase of horizontal resolution to a few kilometers grid spacing to enable an explicit treatment of processes such as precipitating deep convection or gravity waves that need to be parameterized at coarser resolutions (e.g., Satoh et al., 2019). The DYAMOND project (Stevens et al., 2019) was the first intercomparison of simulations of nine of such models which the authors called global storm-resolving. Several of these

models were run with two different horizontal grid spacings to estimate the impact of this configuration feature. Hohenegger et al. (2020) ran simulations with the ICOsahedral Non-hydrostatic (ICON) model for identical setups except for six different





horizontal grid spacings ranging from 2.5 to $80\,\mathrm{km}$ to analyze the convergence of bulk properties of the simulated climate. They show that for many, but not all, important quantities the effect of a halving of the grid spacing decreases with the grid spacing. Top of the atmosphere (TOA) flux differences between the two finest spacings of 5 and $2.5\,\mathrm{km}$ are of about $4\,\mathrm{W\,m^{-2}}$

for outgoing shortwave and less than $1\,\mathrm{W\,m^{-2}}$ for outgoing longwave radiation. Hohenegger et al. (2020) conclude that a "grid spacing of $5\,\mathrm{km}$ appears to be sufficient for capturing the basic properties of the climate system".

  While the DYAMOND models are operated with, in general, more than an order of magnitude finer horizontal grid spacing than typical climate models as participating in the most recent phase of the Coupled Model Intercomparison Project Eyring et al. (CMIP6; 2016), vertical grid spacings used in these two classes of models are rather similar (see Fig. 1.19 of Chen et al.

(2021) for an overview of CMIP model resolutions). The aspect ratio of horizontal to vertical grid spacing is hence much smaller for DYAMOND than for CMIP models. The DYAMOND models extend vertically up to between 37 and $85\,\mathrm{km}$ and deploy between 74 and 137 vertical layers. The potential effect of different choices of vertical grid spacing for the performance of global storm-resolving models (GSRMs) has received less attention than effects of horizontal resolution. The aim of this study is to quantify the role of vertical model grid spacing in global storm-resolving simulations with ICON. More specifically,

the primary goal is to analyze the potential convergence of key climate quantities with the refinement of vertical grid spacing similar to the analysis of Hohenegger et al. (2020) for horizontal grid spacing.

  In traditional global climate modeling, where vertical heat transport and effects of gravity waves are parameterized, the importance of an appropriate aspect ratio of horizontal to vertical model grid spacing has been emphasized in several past studies that typically focus on large-scale dynamical processes. Roeckner et al. (2006), e.g., explored the hypothesis of Lindzen

and Fox-Rabinovitz (1989) whereby appropriate ratios can be estimated based on quasi-geostrophic considerations for large-scale flows and the dissipation conditions for gravity waves. More recently Skamarock et al. (2019) analyzed simulations with a numerical weather prediction model in configurations with horizontal grid spacings down to $3\,\mathrm{km}$ and argued, using dynamical convergence criteria, that a vertical grid spacing of $200\,\mathrm{m}$ or less would be required for convergence. Our study will, however, not concentrate on potential effects on simulated dynamics, but on the global atmospheric energy budget for which the

resolution of clouds is key. Bogenschutz et al. (2021) tested the sensitivity of boundary layer and in particular stratocumulus clouds to vertical grid spacing in a general circulation model (GCM) with $1°$ horizontal resolution but with vertical grids differing mainly in the boundary layer where the maximum layer thickness was between about 15 and $125\,\mathrm{m}$. They showed increasing low cloud amounts with increasing vertical resolution but demonstrated that also variations of the integration time step have an effect. Lee et al. (2022) and Bogenschutz et al. (2023) demonstrated an improvement of simulated stratocumulus

clouds for a refinement of the horizontal grid spacing from $1°$ to $0.25°$ accompanied by a strong refinement of the vertical resolution in the boundary layer for selected processes relevant for the cloud formation.

  We are not the first to ask how the choice of a vertical grid influences the behavior of storm-resolving simulations that explicitly represent vertical heat transport and gravity waves. The focus of such studies is mostly on the interaction of cloud microphysical processes and circulations. Seiki et al. (2015) analyzed simulations with the Nonhydrostatic Icosahedral Atmospheric Model (NICAM) with a minimum horizontal grid spacing of $14\,\mathrm{km}$ and four different vertical grids with between

40 and 236 layers corresponding to maximum tropospheric layer thicknesses between slightly more than $1000\,\mathrm{m}$ and $100\,\mathrm{m}$,





respectively. Their focus was on tropical cirrus clouds. They concluded that a grid spacing of $400\,\mathrm{m}$ or less is necessary for a proper representation of cirrus in their model. Using a more idealized radiative convective equilibrium (RCE) configuration of NICAM, Ohno and Satoh (2018) and Ohno et al. (2019) tested vertical and horizontal grid spacings comparable to those

of Seiki et al. (2015) plus an even finer vertical grid with 398 layers and a maximum tropospheric thickness of $50\,\mathrm{m}$. Their simulations indicate that relative humidity near the tropopause is strongly enhanced for the configuration with the coarsest vertical grid in comparison to all other configurations. Furthermore, ice cloud cover is decreasing and precipitation increasing with a refinement of the vertical grid.

Besides the above mentioned studies with NICAM, there are a number of studies using limited-area and large-eddy models

to estimate effects of vertical grid spacing for the representation of clouds. Using convection-permitting simulations for the region of Northern Africa, Mantsis et al. (2020) argue that their simulation of Saharan mid-level cloudiness depends strongly on vertical resolution and suggested that $50\,\mathrm{m}$ or less might be needed for convergence. As another example, Marchand and Ackerman (2011) note, based on large-eddy simulations (LES), that "accurate [...] simulations of [...] boundary layer clouds are difficult to achieve without vertical grid spacing well below $100\,\mathrm{m}$, especially for inversion-topped stratocumulus." Mellado

et al. (2018) also discuss the representation of stratocumulus clouds in LES. They summarize from earlier studies that a vertical grid spacing of $5\,\mathrm{m}$ near the inversion provides skill for the simulation of the clouds' liquid water path (LWP). Their LES simulations with horizontal grid spacing between $10\,\mathrm{m}$ and $250\,\mathrm{m}$, and vertical spacing between $5\,\mathrm{m}$ and $20\,\mathrm{m}$ show that LWP and low cloud fraction get larger for coarser horizontal and finer vertical grids. Effects of a too coarse horizontal grid on cloudiness may hence be compensated by a too coarse vertical grid.

In our experiments we use the global ICON atmospheric model with grids that have maximum tropospheric vertical grid spacings between 50 and $800\,\mathrm{m}$, i.e. comparable to those used by Ohno and Satoh (2018), but a finer horizontal resolution of, in general, $5\,\mathrm{km}$ and a realistic global configuration. Based on the experience from LES ($\sim$5 to $500\,\mathrm{m}$ horizontal grid spacing) and limited area storm-resolving modeling ($\sim$500 m to $5\,\mathrm{km}$) we have to expect that even the finest vertical grid spacing is too coarse for the realistic simulation of some aspects of global cloudiness. Nevertheless, given the substantial dependence of

cloudiness (and other variables) on the vertical grid spacing reported for a variety of model types, systematically analyzing the effect of much finer vertical grid spacings than typically used in GSRM studies, so far, seems of merit. To compare the potential benefits of refinements in vertical and horizontal grid spacings we have additionally performed a simulation with the same setup but a halved horizontal grid spacing. Moreover, as increases in both horizontal and vertical resolution usually require a decrease of the model's integration time step we have also performed simulations with identical vertical grid spacings

but differing time steps to separate effects of spatial and temporal resolution, or in other words, sensitivities of spatial and temporal truncation errors.

The effect of the choice of an integration time step length on simulation results of global climate models is arguably discussed less frequently than effects of spatial resolution. However, Wan et al. (2021) and Bogenschutz et al. (2021), with the same GCM, demonstrated a strong time step sensitivity of low clouds. In their model that uses convection parameterizations the low cloud

fraction decreases with a shorter time step. Earlier, e.g. Wan et al. (2013) and Gettelman et al. (2015) documented effects of



the length of the integration time step on aerosol and cloud microphysics, and Beljaars et al. (2018) showed an effect on winds in a weather forecast model.

In the following section we present more details on the model and the experimental setup. Section 3 presents effects of a refinement of the vertical grid compared to the reference grid first in terms of global averages, second spatial distributions, and third vertical profiles. Effects of a coarser than standard vertical grid are discussed in Section 4. Section 5 compares effects of changes in horizontal and vertical grid spacings. The final section summarizes the results and provides conclusions.

## 2 Model description and experiment setup

**Table 1.** Key parameters for the simulations.

| Name | # of. layers | max. troposph. layer thickness / m | hor. grid spacing / km | lowest layer thickness / m | integration time step / s |
|------|----------|-----------------|----------------|------------------|---------------|
| L55 | 55 | 800 | 5 | 20 | 40 |
| L110 | 110 | 400 | 5 | 20 | 40 |
| L190 | 190 | 200 | 5 | 20 | 30 (20)* |
| L320 | 320 | 100 | 5 | 20 | 15 (10)* |
| L540 | 540 | 50 | 5 | 20 | 8 (6)* |
| | | | | | |
| L110-2.5km | 110 | 400 | 2.5 | 20 | 20 |
| L55-15s | 55 | 800 | 5 | 40 | 15 |
| L110-15s | 110 | 400 | 5 | 20 | 15 |

*: Time step lengths given in brackets were used for few individual days of the simulations only, where the originally chosen time step caused instabilities.

The numerical model employed in this study is the atmosphere component of the Sapphire configuration of the ICON modeling framework. The Sapphire configuration is described by Hohenegger et al. (2023) and is intended for simulations with horizontal grid spacings of less than $10\,\mathrm{km}$. The atmosphere component uses a dynamical core to solve the nonhydrostatic version of the Navier-Stokes equations and conservation laws for mass and thermal energy and a tracer transport scheme as developed by Gassmann (2011) and Wan et al. (2013), and implemented by Zängl et al. (2015). The only parameterizations of subgrid-scale atmospheric processes used in this model configurations are the treatment of radiant energy transfer, cloud microphysical processes, and turbulent mixing. Radiant energy transfer is calculated with the RTE-RRTMGP scheme (Radiative Transfer for Energetics - RRTM for General circulation model applications, Pincus et al., 2019). Microphysics are parameterized with a one-moment scheme (Baldauf et al., 2011) that was adapted from the numerical weather prediction configuration of ICON (ICON-NWP Zängl et al., 2015). Turbulence is represented through a Smagorinsky-type parameterization





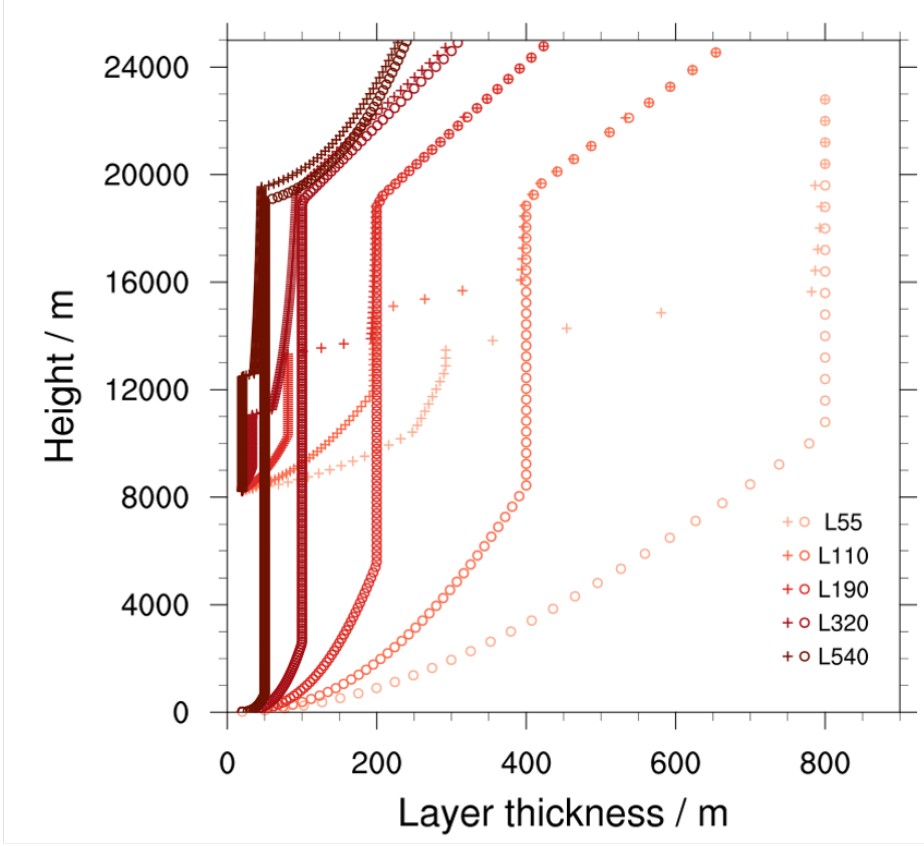

**Figure 1.** Vertical layer thickness for the examples of grid boxes with surface altitudes of 0 m (circles) and 8205 m (crosses), respectively. The y-axis shows the height of the lower edge of the respective layer. Colors indicate the experiments (see Table 1 for details) as indicated in the legend. Darker colors mark finer vertical grid spacing. Only the lowest 25 km of the grids are shown. The model top is at 75 km for all configurations.

(Smagorinsky, 1963) of which the implementation is described by Lee and Hohenegger (2022). Other parameterizations typ-
ically employed for larger-scale atmospheric models to represent the effects of unresolved convection, gravity waves, etc. are
not included in ICON-Sapphire, even though we acknowledge that at storm-resolving scales these are not all well resolved.
Omitting their parameterization allows us to explore their effects as grid spacing is refined, something parameterizations at-
tempt to hide. Land surface processes are treated by version 4 of the Jena Scheme for Biosphere-Atmosphere Coupling in
Hamburg (JSBACH) of which the updates to the earlier version 3.2 (Reick et al., 2021) are described by Hohenegger et al.
(2023). The model equations of ICON-Sapphire are discretized using an icosahedral-triangular C-grid that is described and
illustrated by Giorgetta et al. (2018).

For the vertical discretization, all ICON atmosphere configurations are employing the Smooth LEvel VErtical coordinate
(SLEVE Leuenberger et al., 2010), a terrain-following hybrid sigma z-coordinate. The decay of the effect of topography on





the vertical coordinate is defined using functions described by Leuenberger et al. (2010). Parameters that define the grid for a specific configuration of ICON are in particular the minimum and maximum tropospheric layer thicknesses ($\Delta z_{min}$ and $\Delta z_{max}$, respectively) where the minimum is used for the thickness of the lowest layer, the maximum altitude up to which these limits are applied ($z_{trop}$), and the altitude above which a fixed-height grid is used ($z_{fh}$).

To estimate the sensitivity of the simulated climate to the vertical grid spacing we have performed five simulations with $\Delta z_{max}$ of 50, 100, 200, 400, and $800\,m$ corresponding to grids with in total between 540 and 55 layers. The first five lines of Table 1 show these simulations which are all run at a horizontal grid spacing of about $5\,km$ (measured as the square root of the triangle surfaces; called R2B9 following the terminology in Giorgetta et al. (2018)) and with a model top at $75\,km$. Earlier storm-resolving ICON simulations, e.g. for the DYAMOND project, have often been performed with the same top height and $\Delta z_{max} = 400\,m$, but only 90 vertical layers. Our reference simulation L110 differs from this earlier vertical grid configuration by $z_{trop}$ increased from $14\,km$ to $19\,km$ to have well defined layer thickness changes between the simulations for an altitude region including the tropical tropopause layer. Fig. 1 shows layer thicknesses from the surface to $25\,km$ altitude for the five vertical grids for the examples of surface altitudes of $0\,m$ and $8204\,m$. The latter corresponds to the grid box with the highest surface elevation of an R2B10 grid (with a horizontal grid spacing of about $2.5\,km$). Test simulations not further discussed in this paper had revealed the tendency of the grids with 320 and 540 layers to frequently produce numerical instabilities. This issue could be solved by increasing the altitude where the grid is transitioning from terrain-following to fixed-height, which is why the three coarsest vertical resolutions are run with $z_{fh} = 22\,km$ while for L320 and L540 we are using values of 27 and $35\,km$, respectively.

Four further sensitivity simulations that have been performed are listed in Table 1. To put the effects of changes in vertical grid spacing into perspective we have performed simulation L110-2.5km which differs from the reference simulation L110 by its halved horizontal grid spacing and a halved integration time step. The standard simulations with different vertical (and horizontal) grid spacings use integration time steps of different lengths between $40\,s$ for L55 and L110, and $8\,s$ for L540. To disentangle potential effects of changes in vertical and temporal discretization we have performed the two additional simulations L55-15s and L110-15s with configurations almost[1] identical to simulations L55 and L110, respectively, but a time step of $15\,s$ which is also employed in the L320 experiment. This creates a series of three experiments with identical time step length but still different vertical grid spacing. Simulations L55-15s and L110-15s have the additional benefit that differences to their respective references L55 and L110 can be attributed unambiguously to the different time steps.

The simulations in this study are initialized from the global meteorological analysis taken from the European Centre for Medium Range Weather Forecasts (ECMWF) for June, 27, 2021, 0 UTC, and run for 45 days with daily sea surface temperatures and sea ice fraction taken from ECMWF operational daily analyses as lower boundary condition. Most comparisons of the climate state simulated in the different experiments are performed for the last 40 days of the simulation. Additionally, to enable some estimate of the internal variability of the simulations, four subperiods of 10 days each have been compared for parts of the analysis.

---

[1]Note that L55-15s is the only simulation with a lowest layer thickness of $40\,m$ instead of $20\,m$ used for all other simulations. However, comparison of an earlier L55 simulation with $\Delta z_{min} = 40\,m$ shows negligible effects of this choice in comparison to the L55 simulation used here.





## 3 Effects of vertical grid refinement

In most comparisons we use the simulation L110 as reference because its vertical grid is closest to the standard configuration of ICON-Sapphire. In this section we compare simulations with maximum tropospheric grid spacings reduced by up to a factor
of 8 (L540) with respect to this reference. The simulation with the coarsest vertical grid spacing (L55) is presented separately in Section 4 because it behaves differently in several respects than one might extrapolate from the experiments with finer vertical grid spacing. Section 5 compares effects of horizontal and vertical grid refinements. The first two subsections of this section report model results in terms of global means and horizontal patterns. The presentation of effects on vertical profiles in Section 3.3 is used to discuss potential reasons for the simulated effects. This discussion is continued in Sections 4 and  5.

### 3.1 Global averages

#### 3.1.1 Combined effects of vertical grid spacing and time step changes

When refining the spatial discretization of a numerical general circulation model it is, in general, necessary to decrease the integration time step to ensure the stability of the integration by avoiding the violation of the Courant–Friedrichs–Lewy (CFL) convergence condition. Therefore, we are first discussing the effect of combined changes of the vertical grid spacing and the
integration time step for the first five experiments listed in Table 1. For simplicity, we refer to these combined changes as "practical vertical resolution changes" because the goal is to change the vertical grid spacing and the time step is changed out of necessity. Individual effects of grid spacing and time step changes will be presented in Subsection 3.1.2.

Figure 2 a) shows differences in global mean precipitation, precipitable water, and vertically integrated cloud ice and liquid water of experiments L190, L320, and L540, respectively, compared to the reference experiment L110. Experiments L55 and
L110-2.5km are included in this figure but will be discussed in the following sections.

Precipitation differences among the experiments are small and, as indicated by the vertical bars, can be of different sign for the same experiment but different 10-day subperiods. Possibly, 10-day simulation periods are too short for unambiguously representing a potential effect of vertical grid spacing on precipitation. Over longer time spans precipitation is constrained by the atmospheric energy balance (e.g., Mitchell et al., 1987; Fläschner et al., 2016). Summing up the radiation fluxes of Fig. 2 d)
and g) indicates increases of atmospheric radiative cooling of between about $1\,\mathrm{W\,m^{-2}}$ for the practical refinement to L190 and $3\,\mathrm{W\,m^{-2}}$ for the practical refinement to L540. These increases are largely balanced by the turbulent energy fluxes at the surface and hence consistent with small increases of precipitation. The energy fluxes will be discussed in more detail below. Precipitable water decreases with practical increases of vertical resolution. The difference between L540 and L110 is of about $2\,\%$.

A large dependence on a practical vertical grid refinement is simulated for vertically integrated cloud liquid water which increases by similar amounts for each halving of grid spacing. The effect reaches about 22 % for the reduction of the maximum tropospheric grid spacing from $400\,\mathrm{m}$ in L110 to $50\,\mathrm{m}$ in L540. There is no sign of convergence within the range of grid spacings used here. Vertically integrated cloud ice also increases with vertical resolution, but the effect is smaller than for





liquid water. The L190, L320 and L540 experiments show increases in vertically integrated cloud ice of about 5, 7, and 8 %
compared to L110, which indicates convergence for this quantity.

Effects on global mean vertical energy fluxes at the top of the atmosphere (TOA) and the surface are shown in Figs. 2 d) and
g), respectively. Net downward TOA shortwave radiation decreases with practicallly increasing vertical resolution (by about
$2.5\,\mathrm{W\,m^{-2}}$ for L540 compared to L110) as it may be expected from the increase of cloud water. Outgoing longwave radiation
(OLR) seems to increase with practically increasing vertical resolution, although the spread between the individual 10-day
periods is large.[2] From the increase of cloud ice one would a priori expect the opposite behaviour, but the sign is consistent
with the reduction of precipitable water. The apparent inconsistency of the cloud ice and longwave radiation effects will be
discussed in Section 3.3.

The effects on the net shortwave downward radiation at the surface are, for all experiments, similar to its TOA component.
Net surface downward longwave radiation is increasing with practical increases of vertical resolution which would be consistent
with more cloud liquid water and ice. The upward sensible heat flux is increasing with practical increases of vertical resolution,
as it also seems to be the case for the latent heat flux. The latter signal is very variable and may not be robust but is consistent
with increasing atmospheric radiative cooling mentioned above.

In summary, the individual components of the TOA and surface energy fluxes presented in Fig. 2 respond very differently in
magnitude to practical changes of vertical resolution, but none of them shows a clear tendency of convergence for the range of
vertical grid spacings tested here.

Hohenegger et al. (2020) compared the effects of their horizontal resolution changes in ICON to the multi-model standard
deviation of the DYAMOND model intercomparison project. In terms of global means they concluded that simulated differ-
ences between horizontal grid spacings of 2.5 and $5\,\mathrm{km}$ are small in comparison to the multi-model spread. As we will show
in Section 5 effects of halving the vertical grid spacing are in general smaller than of halving horizontal grid spacing. Hence,
also the effects of practical vertical resolution doublings are small with respect to this metric.

In the following, we will first analyze to which degree the reported changes in selected global mean climate quantities are
due to changes in spatial resolution or time step length. Following this, we will analyze horizontal and vertical patterns of the
changes to better understand the origin of global mean effects. The focus will be on the robust signals in both cloud condensate
components and their effects on radiative fluxes.

### 3.1.2 Individual effects of vertical grid spacing and time step changes

Panels b), e), and h) of Fig. 2 show differences in the same quantities discussed above of experiment L320 to L110-15s. These
two experiments use the same integration time step of $15\,\mathrm{s}$ (except for a single day of L320 that was run with $10\,\mathrm{s}$ to overcome
an instability) but differ in their vertical grid spacing. Panels c), f), and i) of Fig. 2 show differences of experiments L110-15s
and L110, and of experiments L55-15s and L55, i.e. the effects of a pure reduction of the integration time step from 40 to $15\,\mathrm{s}$.
Comparison of Panels a), b), and c) of Fig. 2 indicates that the increase of cloud liquid water is due to both changing vertical
resolution and time step. Both changes appear to contribute about equally to the total effect although the exact contributions

---

[2]Note that for consistency all fluxes, including longwave radiation, latent and sensible heat, are presented as positive downward in the figures.



are uncertain due to temporal variability. While in L320 the maximum tropospheric vertical grid spacing is reduced to 1/4 of the L110 value the time step is reduced only to 3/8.

An increase in cloud liquid water would be consistent with an increase in reflected shortwave radiation as mentioned above. However, the previously discussed reduction of TOA downward shortwave radiation with increasing vertical resolution seems to be largely a time step effect while the pure resolution effect has little influence.

In contrast to cloud liquid water, the effect on vertically integrated cloud ice reported above is dominantly indeed a vertical resolution dependence. The reported effect on the sensible heat flux, on the other hand, is largely a time step effect.

The global net radiative effect of a practical vertical resolution increase is a cooling. In experiment L320, e.g., in comparison to L110, the global total net downward energy fluxes at both TOA and surface are smaller by about $1.9\,\mathrm{W\,m^{-2}}$. Both effects are dominated by the time step dependence. The origin of this will be discussed below.

## 3.2 Spatial distributions

Fig. 3 shows zonal mean differences of selected cloud and radiation flux quantities between selected simulations. To facilitate the identification of signals we don't present results from all practical vertical refinement experiments but leave out experiment L190, which causes similar but weaker signals than the even finer resolved grids, and limit ourselves to the differences (L320 - L110-15s) for the pure resolution effect and (L110-15s - L110) for the pure time step effect. The zonal mean differences (L320 - L110) and (L540 - L110) show similar latitude dependence for all presented quantities with in general larger amplitudes for the latter difference. This increases our confidence in the robustness of these signals.

Fig 3 a) indicates that the global increase of vertically integrated cloud liquid water with a practical increase of vertical resolution is due to increases over a large latitude range from southern to northern mid-latitudes. As for the global mean differences there is also no tendency for convergence with higher resolution over all this latitude range. Despite some variability the curves also confirm that the pure time step effect (L110-15s - L110) has a larger contribution than the pure vertical resolution effect (L320 - L110-15s) to the practical resolution effect (L320 - L110). While there appears to be a distinct pattern also at higher, in particular northern, latitudes, this pattern is much more dependent on the averaging period than the effects at lower latitudes and may depend strongly on the accidental evolution of the meteorological situation in the different simulations. In the analysis of the vertical structure of effects of resolution changes (Section 3.3) we will therefore concentrate on tropical (30°S to 30°N) averages. The difference maps of Fig. 4 confirm that the increase of cloud liquid water is widespread over middle to low-latitude oceans and that both the time step and the vertical grid spacing contribute to this effect. Based on past studies, one may expect that in particular the amount of liquid water in stratocumulus areas increases for finer vertical grid spacing. Indeed, the patterns of the difference (L320 - L110) shown in Fig. 4 and of (L540 - L110) (not shown) indicate a relatively strong increase off the west coast of northern South America but this signal certainly does not dominate the zonal mean and is not visible in other stratocumulus regions as, e.g., off the west coast of North America. Furthermore, it seems again that the time step effect, as indicated by the difference (L110-15s - L110) contributes strongly to the South Pacific stratocumulus signal.

The latitudinal distribution of the increase in vertically integrated cloud ice (Fig. 3 b) is much less uniform than that of liquid water, the same is true for the spatial pattern shown in Fig. 5. While this can't be easily identified from the map, in





Section 3.3.2 it will be shown that the signal differs on average between moist and dry regions of the tropics. In the zonal mean, pure and practical increases of vertical resolution lead to increases of the zonal mean cloud ice amount with peaks near 40°S, 10°S, 20°N and 50°N. It is unclear to which degree these peaks are due to the accidental evolution of weather in the individual experiments or actually a signal of the resolution differences. Zonal average differences calculated for the first 5 days of the simulation period (not shown), for which the simulated meteorology is still fairly similar among the simulations, indicate more uniform increases for finer vertical grid spacings but in particular very similar effects of the practical increase to 320 layers and the pure effect of such a resolution increase. This confirms that the ice signal is, different than the liquid water signal, largely unaffected by the time step differences. In contrast, the local increase of cloud ice over northern South America with a practical resolution increase is mostly a time step effect.

The effect of increasing vertical resolution on cloud liquid water is strongly correlated with the effect on zonal mean outgoing, i.e. reflected, shortwave radiation (compare Figs. 3 a) and d)) and anti-correlated with net surface shortwave radiation (not shown). Regional maxima of an increase in shortwave reflection are for example simulated in the above mentioned stratocumulus region off South America with values up to about $30\,\mathrm{W\,m^{-2}}$ for the comparison (L540 - L110). The correlation breaks down near the intertropical convergence zone (ITCZ) which is centered slightly north of the equator during the simulation period and where no increase of outgoing shortwave reflection is simulated despite increases in cloud liquid water. Here, ice clouds and their changes dominate the changes in reflection. Changes in outgoing shortwave radiation are, in general, well correlated with changes in total cloud fraction diagnosed in the simulations.

## 3.3 Vertical profiles

### 3.3.1 The boundary layer

We have mentioned above that vertically integrated cloud liquid water increases over a large latitude range dominantly due to a reduction of the time step but additionally due to an increase in vertical resolution. Fig. 6 shows the vertical profile of cloud liquid water averaged over 30°S to 30°N and the differences in this quantity due to individual and combined changes of grid spacing and integration time step. It is clear that the different changes have distinct effects on the profile.

A shorter time step increases the liquid water content in the cloud layer, and dominantly near the lifting condensation level. A higher vertical resolution shifts the profile upwards. While time step and resolution effects are not necessarily independent of each other (see Section 4) the effect of a practical vertical resolution increase is, in general, well approximated by a combination of the two individual effects. It is clear from Fig. 6 that the effect of increased vertical resolution on cloud liquid water is not converging in the range of vertical resolutions used in these experiments.

The origin of the increase in cloud liquid water for practically increasing vertical resolution is not easy to identify. Bogenschutz et al. (2021), and earlier Wan et al. (2021) with the same GCM also recognized a strong time step sensitivity of low clouds. However, in their model that uses convection parameterizations the low cloud fraction decreases with a shorter time step. Wan et al. (2021) show that this sensitivity depends on specifics of the numerical coupling of various processes in the model integration, in particular the parameterizations of convection and cloud micro- and macrophysics.





Concerning the effect of a spatial refinement one may assume that when splitting a grid box in two there is simply a larger chance to reach saturation. While we can't exclude such an effect other processes seem to dominate because a refinement of the vertical grid produces an altitude shift, and a refinement of the horizontal grid even a reduction of cloud liquid water (see Section 5).

To better understand the effects we will have a closer look at marine stratocumulus off the coast of South America where signals are strong (Fig. 4). As mentioned earlier, an increase of cloud liquid water in the marine stratocumulus regions may be expected for a finer vertical grid spacing due to a potentially better resolution of the inversion layer, and is also simulated in our experiments. Fig. 7 shows vertical profiles of cloud liquid water, temperature, and specific humidity averaged over the region $15°$S to $0°$N and $105°$W to $90°$W. The profiles support the interpretation of a deepening boundary layer and an upward shifting cloud layer for finer vertical grid spacing (Fig. 7b). The upward shift of the inversion can be seen in the temperature profile that shows a reduction of almost $2\,\mathrm{K}$ near $1.8\,\mathrm{km}$ when comparing L320 and L110. The profiles of cloud liquid water, specific humidity, and relative humidity (not shown) all show a strong reduction near the reference cloud base and a strong increase near the reference cloud top. Apparently, the better representation of the inversion leads to a deepening of the boundary layer and also a drying of the free troposphere (see Fig. 7f).

In this stratocumulus region, the individual effects of vertical resolution and time step changes are less shifted against each other in altitude than over the average tropics and both contribute to the liquid water increase near the reference cloud top. The overall increase is mainly due to the time step effect because it is basically positive all over the cloud layer while the pure resolution effect is a vertical shift. However, in contrast to the tropical average, the simulated changes to the reference don't increase much when going to even finer grid spacings than in L190 (not shown). This occurs despite the experience from LES simulations of a large sensitivity to changes in vertical grid spacings even at much smaller scales than considered in our simulations (e.g., Marchand and Ackerman, 2011; Mellado et al., 2018).

An increase of stratocumulus cloud amount was also simulated in a GCM by Bogenschutz et al. (2021) for an up to 8-times increase of the number of vertical levels in the boundary layer. However, only in the first two doublings the boundary layer depth increased as in our case. The last doubling led to a decrease of the depth. Moreover, in their case a shorter integration time step led to a reduction of the cloud amount.

While the effects of resolution changes in the selected stratocumulus region are much larger than in the tropical average there are a lot of qualitative similarities which suggests that similar mechanisms are in play all over the tropics. This widespread occurrence of similar changes is confirmed by Fig. 8 which shows changes of tropical vertical profiles of cloud liquid water in precipitation space. The upward shift caused by the finer grid spacing and the increase of cloud liquid water dominantly in the lower part of the cloud layer for a shorter time step is happening in most precipitation regimes.

Cloud liquid water effects averaged over southern mid-latitudes also look similar, while the largest cloud liquid water amounts in the northern hemisphere occur in the Northern Pacific and Atlantic regions (Fig. 4) in a very shallow boundary layer, and the resolution effects are regionally varying. It is plausible that a change of the ventilation of the boundary layer and possibly also a reduction of numerical diffusion through the refinement of the grid contribute to the changes of cloud liquid water simulated over such a variety of different conditions.





### 3.3.2 The free troposphere

We reported above that vertically integrated cloud ice is hardly affected by changes in the integration time step but almost only by the pure changes of vertical grid spacing. Fig. 9 shows that refining the grid spacing basically increases tropical cloud ice at all altitudes where it also exists in the reference simulation. The largest absolute increase is simulated near the cloud ice peak at around $11\,km$. Only near $14\,km$ cloud ice is slightly reduced for finer vertical grid spacing. The figure also confirms the earlier notion that the effect of refining the grid spacing seems to converge for the highest resolution used in our simulations.

To better understand possible causes and consequences of this change in cloud ice Figs. 10 and 11 show tropical mean vertical reference (L110) profiles of temperature, specific humidity, vertical wind, relative humidity, cloud fraction, and upward longwave radiation, and their respective differences for selected experiments. As an indicator for convective activity the vertical wind velocity is not averaged over the whole tropics but over the 10% of the tropical area with the strongest precipitation for a given time interval.[3]

Cloud ice increases with a refinement of the vertical grid (Fig. 9c) although specific humidity (Fig. 10d) and, despite the lower temperature (Fig. 10b), also relative humidity (Fig. 11b) are reduced at almost all altitudes where cloud ice exists.

Fig. 12a shows tropical cloud ice profiles in precipitation space and their differences between experiments L540 and L110.[4] The average increase of cloud ice is dominated by the regions with high precipitation, i.e. the regions where deep convection occurs. In tropical regions with low precipitation the vertically integrated ice cloud content (not shown) decreases by about 335 20% which means that in these regions the changes in relative humidity and cloud ice are consistent, but this is not the case in the regions with high precipitation. There, the increase of cloud ice is likely related to the increase of the upwelling (Fig. 10f) transporting more condensate upwards.

The increase of upwelling is consistent with the drying of the troposphere, visible in both specific and relative humidity (Figs. 10d and 11b)), which increases atmospheric radiative cooling that needs to be balanced by stronger subsidence in the 340 dry tropical regions. The effect on atmospheric moisture will be discussed in the following section because it is particularly strong for a coarsening of the vertical grid. The change of upwelling in the moist tropics is also consistent with the temperature changes. A pure time step decrease (L110-15s - L110) reduces temperatures in the boundary layer and convective upwelling. A pure vertical resolution increase (L320 - L110-15s) decreases temperature over almost all the tropical profile above the boundary layer, and thereby increases the convective available potential energy and upwelling. This combines to an almost 345 unchanged upwelling up to about $8\,km$ and an increase up to about $14\,km$ for the practical increase of vertical resolution (L320 - L110).

Fig. 12b shows that cloud fraction is reduced everywhere in precipitation space at tropical ice cloud altitudes, despite the increase of the cloud ice content in high precipitation areas. Thus, the opposing signals in cloud ice concentration and cloud fraction (Figs. 9 and 11d) are dominated by the changes in the areas of high precipitation, and not the subsidence regions where

---

[3]The magnitude of this average upwelling depends strongly on the selection of the precipitation threshold and on the time average over which this threshold is applied (here daily), but the shape of its vertical profile is fairly independent of these choices.

[4]The pattern of the difference to the reference is consistent for all experiments with a finer vertical grid spacing and also similar if all tropics or only tropical ocean areas are considered.





the changes have the same sign. ICON-Sapphire uses a cloud scheme that sets cloud fraction in individual grid boxes to 1 if the total condensate concentration is above the threshold of $10^{-3}\,\mathrm{g\,kg^{-1}}$, else to 0. A lower average cloud fraction hence means that fewer grid boxes are filled with a sufficient condensate mixing ratio. The apparent contradiction of lower cloud fraction but higher cloud ice concentration at some altitudes can only be explained when there is a larger concentration in fewer condensate-filled grid boxes. This would be consistent with the distribution of lower supersaturation over larger grid boxes suggested by

Seiki et al. (2015) for coarser vertical grid spacing. They argued that finer vertical grid spacing would accelerate the interaction between clouds and radiation due to less air mass in a cloudy grid box and hence faster temperature change. Increased cloud top cooling in such boxes would decrease stability and increase variability of supersaturation. However, it is not clear if this the dominant process in our simulations. Consistent with our results, also Ohno et al. (2019) simulate a reduction of high-cloud fraction for finer vertical grids in RCE simulations with NICAM at $28\,\mathrm{km}$ horizontal resolution.

## 4   Effects of vertical grid coarsening

Fig. 2 shows that many global mean quantities change more strongly for coarsening the vertical grid with respect to the reference (L55 - L110) than for refining it. In the case of precipitation, for instance, the (L55 - L110) and (L55-15s - L110-15s) differences are the only signals for which the spread between the four individual 10-day periods does not include zero. Both differences result purely from the change in vertical grid spacing because the compared simulations use identical time steps of

$40\,\mathrm{s}$ and $15\,\mathrm{s}$, respectively. The similarity of the differences (L55 - L110) and (L55-15s - L110-15s) provides confidence in the robustness of the signals.

The decrease of precipitation of about 4% is consistent with the reduction of the latent heat flux and with the reduction of the atmospheric radiative cooling, which is clearly visible in the reduction of OLR. In the case of (L55 - L110) OLR is reduced for the coarser vertical grid spacing by about $4.3\,\mathrm{W\,m^{-2}}$, in the case of (L55-15s - L110-15s) only by about $2.6\,\mathrm{W\,m^{-2}}$. Although

the spreads are overlapping we can't exclude that the effect of the change in vertical grid-spacing has a time-step dependence. However, even the $2.6\,\mathrm{W\,m^{-2}}$ effect is larger than any of the effects simulated for a refinement of the vertical or horizontal grids and merits further attention.

Fig. 13 shows that the reduction of OLR is dominated by a tropical signal and consistent with an increase in cloud fraction that reaches a maximum of about 12 percentage points near 10°S. Both signals are larger than signals of even the strongest

refinement of the vertical grid (L540 - L110). By contrast, signals from a coarsening of the vertical grid spacing in cloud ice are relatively small, consistent with the global mean quantities. Due to the dominance of the tropical signals it makes sense to, again, discuss vertical profiles of quantities averaged over 30°S to 30°N.

Similar to several global mean quantities (Fig. 2), experiment L55 also stands out in terms of the tropical profiles of, for instance, temperature and specific humidity because the differences to the reference are larger than simulated in all experiments

with refined vertical grid spacing. Temperature increases all over the troposphere with a maximum of about $1.3\,\mathrm{K}$ reached near $14\,\mathrm{km}$, compared to a maximum decrease under pure vertical grid refinement (L320 - L110-15s) of less than $0.5\,\mathrm{K}$. Specific humidity increases up to 50% in the upper troposphere for a halving of vertical resolution compared to a maximum decrease



of about 10% for quadrupling it. Also relative humidity (Fig. 11) increases much more strongly, by more than 10 percentage points, in the upper troposphere when comparing L55 to L110 than it decreases for increasing resolution. Opposite to the mois-
ture decrease discussed above for the refinement case the strong moisture increase for the coarsening would reduce atmospheric radiative cooling and hence decelerate the overturning circulation. Lang et al. (2023) analyzed moisture differences in the tropical middle troposphere in our simulations L55, L110, L190, L110-2.5km, and in further simulations with parameter changes. In general, the largest contribution to differences between different model configuration can be traced back to differences in the conditions an air parcel experienced at the point of its last saturation. However, in the case of increased moisture due to the
coarsening of the vertical grid spacing, about half of the increase can't be explained by this. Lang et al. (2023) speculate that an increase of numerical diffusion for coarser grids contributes to this. Similar to our results, Ohno et al. (2019) also simulate the strongest increase of tropospheric humidity when switching to their coarsest vertical resolution with grid spacings in the upper troposphere of about $1\,\text{km}$.

While the refinement experiments cause increases of cloud ice at almost all altitudes, the signal of experiment L55 is
characterized by an upward shift (Fig. 9) of the profile. Also this signal may be caused by the strongly increased moisture which shifts the maximum divergence of longwave radiative cooling and thereby the divergence of vertical velocity in downwelling areas of the tropics and the convective anvils upward (e.g., Hartmann and Larson, 2002). In precipitation space (not shown) the signal of the coarsening of the grid spacing is qualitatively opposite to the signal of the refinement shown in Fig. 12. However, with the relatively strong increase of relative humidity cloud ice also increases strongly in the dry regions. At altitudes near
$13\,\text{km}$ this overcompensates the reduction through reduced upwelling in the moist regions.

The dipole in the upper tropospheric temperature anomaly in this simulation translates itself into an upward shift of the cold point tropopause of about $1\,\text{km}$ but hardly any change of the cold point temperature.

As mentioned above, a remarkable feature of the L55 simulation is its strongly reduced OLR (by more than $4\,\text{W}\,\text{m}^{-2}$ globally and more than $6\,\text{W}\,\text{m}^{-2}$ in the tropical average; Figs. 2d and 11f, respectively). The analysis of clear-sky fluxes (not
shown) indicates that about half of this is a clear-sky effect related to the strongly increased tropospheric humidity (Fig. 10d). The other half can be explained by the increase in ice cloud fraction (Fig. 11d). Analogous to the refinement experiment where the ice cloud fraction is reduced everywhere in precipitation space the coarsening leads to an increase everywhere (not shown). In the rainy regions this increase occurs despite a decrease of the cloud ice amount. As discussed above, this apparent inconsistency in the moist regions can only be explained by less cloud ice being distributed to a larger number of grid cells.

Seiki et al. (2015) analyzed tropical cirrus in NICAM simulations at 14 and $28\,\text{km}$ horizontal grid spacing for different vertical grid spacings of about 100, 200, 400, and $1000\,\text{m}$ in the upper tropical troposphere, i.e. comparable to our experiments L320 to L55. In their simulations, differences in ice cloud quantities between the three finest grids are also much smaller than for the coarsest grid. Their and our results further agree on a comparable decrease of OLR caused by the coarsest vertical grid spacing. However, in other aspects the results differ. While our cloud fraction increases for the coarsest grid basically at all
upper tropospheric altitudes they report an increase only near the tropopause.





# 5 Comparing effects of horizontal and vertical grid refinements

Panels a), d), and g) of Fig. 2 show the differences in globally averaged water quantities and radiation fluxes of experiment L110-2.5km with respect to the reference experiment L110, i.e. the effects of halving the horizontal grid spacing, in comparison to the differences of the experiments with refined vertical grids discussed above. The signs and magnitudes of the effects of the

practical increase of horizontal resolution presented here are, in general, in agreement with the effects reported and discussed by Hohenegger et al. (2020). Here, the purpose of showing horizontal resolution effects is to compare them with vertical resolution effects. To estimate the relative importance of a practical doubling of horizontal to a practical doubling of vertical resolution one may compare (L110-2.5km - L110) with (L190 - L110). However, to judge the effects of comparable computing time investments in horizontal or vertical refinement the comparison of (L110-2.5km - L110) with (L320 - L110) would be

more appropriate. Independent of which vertical resolution experiment is chosen, absolute effects of horizontal refinement are larger for precipitable water and shortwave radiation fluxes both at the TOA and the surface. Effects on longwave fluxes are of comparable magnitude. The absolute effect on cloud liquid water reaches a similar magnitude when the L320 experiment is used in the comparison, and the effect on cloud ice is larger in the vertical resolution experiments.

Robust (in the sense of the spread of the four individual subperiods not including zero) effects of horizontal and vertical

refinements have different signs in particular for cloud liquid water, shortwave fluxes, and the longwave surface flux. The signs of the longwave TOA flux differences are also opposite, but less robust. By contrast, the effect on cloud ice seems to point in the same direction, but is less robust for horizontal refinement.

In the following we will further discuss the relatively large effects on cloud condensate, and in particular cloud liquid water, which is reduced by about 16 % in experiment L110-2.5km compared to L110. As these experiments differ not only in the

horizontal grid spacings of about 2.5 and $5\,km$, but also in the time steps of 20 and $40\,s$, the time step effect could partly compensate the horizontal resolution effect. If one assumes that the time step effect acts independently of spatial resolution changes, the pure effect of a doubling of horizontal resolution could be larger by 5 %. This suggests that necessary time step adaptations may reduce the effect of changing the horizontal but enhance the effect of changing the vertical grid.

The effects of practical vertical and horizontal resolution changes on net TOA shortwave radiation have opposite signs,

which is consistent with the cloud liquid water changes. However, the shortwave radiative effect of horizontal refinement per unit change of cloud water is larger than for vertical resolution.

Fig. 6 shows that the reduction of cloud liquid water with a practical increase of horizontal resolution is occurring fairly homogeneously all over the vertical profile in the tropics. As a shorter time step causes a qualitatively opposite effect one can assume that the effect of a pure refinement of the horizontal grid spacing would cause an even larger but qualitatively similar

reduction of cloud liquid water. The reduction of low clouds with higher horizontal resolution has been reported earlier in storm-resolving model simulations with ICON (Hohenegger et al., 2020) and NICAM (Noda et al., 2010). In contrast to the ICON simulations performed for this study and by Hohenegger et al. (2020), Noda et al. (2010) used a subgrid-scale shallow cumulus parameterization. They show that increasing the ventilation of the boundary layer through parameter changes of this parameterization also decreases the low cloud amount. This likely happens in ICON simulations through a better resolution



of shallow cumulus for finer horizontal grid spacing. Stevens et al. (2020) argue that a further refinement of horizontal grid spacing to the hectometer scale could further improve such and other cloud features.

Fig. 7 shows that in the South American stratocumulus region the changes in cloud liquid water caused by a halving of horizontal grid spacing are of a similar magnitude as those caused by changed vertical grid spacing despite much weaker effects on the structure of the boundary layer as characterized by the profiles of temperature and humidity. We conjecture that

the clouds are more directly affected by a change in horizontal grid spacing, which in this GSRM is still inadequately large for resolving shallow cumulus, but more indirectly by a change in vertical grid spacing via a modified representation of the inversion.

In terms of the vertical distribution of cloud ice (Fig. 9) a refinement of the horizontal grid spacing has a comparable effect to refining the vertical grid but leads to an increase even at very high altitudes up to about $17\,\mathrm{km}$. Fig. 10 shows that, similar to

the vertical refinement effect, this is likely related to the increased convective activity as indicated by the increased upwelling in moist regions.

## 6 Summary and conclusions

The aim of this study is to quantify the effects that choices of different vertical grid spacings have on the global climate simulated with the GSRM ICON-Sapphire at $5\,\mathrm{km}$ horizontal grid spacing. We have analyzed 40 days simulated with boundary

conditions for a period in the boreal summer of 2021 for five different vertical grids having between 55 and 540 vertical layers and maximum tropospheric grid spacings between 800 and $50\,\mathrm{m}$. The configuration with $400\,\mathrm{m}$ maximum tropospheric spacing is close to the vertical grid typically used in ICON-Sapphire simulations (e.g., Hohenegger et al., 2023). While, in general, simulations with finer grids were run with shorter integration time steps to ensure numerical stability, additional simulations were performed with three different vertical grids but identical time step length. This enables us to disentangle pure effects

of changes in vertical grid spacing and the choice of the integration time step, and to compare them to effects of practical resolution changes which are resulting in general from a combination of both. Furthermore, we have run a simulation with halved horizontal grid spacing to compare effects of vertical and horizontal grid choices.

Practically increasing vertical resolution leads to changes in the global energy budget. Upward shortwave radiation at the TOA increases by about $2.5\,\mathrm{W\,m^{-2}}$ when increasing the number of vertical layers from 110 to 540. The effect doesn't show

any convergence for the vertical grids chosen here and can be attributed mostly to changes in low clouds. Globally averaged cloud liquid water increases by about 7% for every practical doubling of the vertical resolution.

One may expect that the increase of cloud liquid water with finer vertical grid spacing should be particularly strong in stratocumulus regions where the simulation of the inversion at the cloud top likely benefits from finer grid spacing. Such an effect is simulated in our experiments for stratocumulus in the Pacific off South America, but the signal is not robust

over stratocumulus regions in general. Moreover, in the global average, more than half of the effects on cloud liquid water and shortwave radiation are due to the reduction of the integration time step necessary for a refinement of the grid spacing. Bogenschutz et al. (2021) also simulated an increase in the low cloud amount for finer vertical grid spacing but in a GCM with





more traditional horizontal grid spacing that includes a convection parameterization. Contrary to our results, Bogenschutz et al. (2021) simulated a reduction of low cloud amount for a shorter time step. Wan et al. (2021) show that this sensitivity depends

on specifics of the numerical coupling of various processes in the model integration. To better understand and possibly reduce the time step effect in our model would require a more detailed analysis of the numerical integration scheme. However, our results confirm that when changes in model grid spacing require changing the integration time step, effects can not necessarily be unambiguously attributed to the spatial discretization change. On a more general note, this serves as a reminder that, while often the shortest integration time step that avoids model crashes is chosen for practical reasons, this choice can affect the

simulation results strongly.

Refining the vertical grid spacing increases OLR in our experiments. This effect is, however, in general much smaller than the effect on shortwave radiation. The OLR response can be traced back partly to changes in tropical cirrus clouds of which, in the tropical average, the cloud fraction decreases with resolution while the total ice mass increases. Effects on the cloud ice concentration are different in moist tropical regions, where they are likely related to changes in convective activity, and dry

tropical regions, where they are consistent with changes in background conditions. Different than for liquid clouds, effects on tropical upper tropospheric ice clouds and OLR show a tendency to converge for the vertical resolutions tested here. A further difference is that these effects on cloud ice and OLR are dominated by the effect of the vertical grid spacing and negligible influence from time step changes.

By far the largest change in cloud ice is simulated for increasing the maximum tropospheric vertical grid spacing from

$400\,\mathrm{m}$ in the reference configuration L110 to $800\,\mathrm{m}$ in L55. This decreases global OLR by more than $4\,\mathrm{W\,m^{-2}}$. The coarsest resolution differs largely from the other simulations in several other quantities like tropical tropospheric temperature, specific, and relative humidity. A similarly non-linear response to vertical resolution changes was simulated in the NICAM GSRM by Seiki et al. (2015) and a NICAM RCE configuration by Ohno et al. (2019). Seiki et al. (2015) concluded from this that to accurately simulate tropical cirrus clouds a vertical grid spacing of $400\,\mathrm{m}$ or less is required. Our simulations support the

conclusion that vertical grid spacing larger than $400\,\mathrm{m}$ would be inadequate for the simulation of tropical cirrus. More in general, the large effects simulated in ICON-Sapphire for a coarsening of the vertical grid spacing indicate a high price for the computing time gains that could be obtained in such a configuration.

For most climate quantities studied here, doubling vertical resolution beyond the reference grid L110 has smaller effects than the doubling of horizontal resolution to $2.5\,\mathrm{km}$ grid spacing. As an example, cloud liquid water increases by about 7% per

practical doubling of vertical resolution, compared to an approximately 16% decrease for practical the doubling of horizontal resolution. Interestingly, the effects have opposite sign. Effects of increases in horizontal resolution on this quantity are hence counteracted by increases in vertical resolution. Such opposite sensitivities were also obtained by Mellado et al. (2018) in LES simulations of inversion-topped stratocumulus clouds. Our results confirm the importance of the aspect ratio of vertical and horizontal resolution for the simulation of boundary layer clouds also in GSRMs. More in general, our study emphasizes that

the simulation of boundary layer clouds and associated effects in GSRMs is more susceptible to truncation errors than the simulation of many other quantities including, for instance, cirrus clouds.



In our analysis we have concentrated on sensitivities of the simulated climate to resolution changes, and not on the question if changes in vertical resolution lead to a better representation of certain climate features in comparison to observations. The latter question is of course of interest. However, with our focus on globally or tropically averaged climate quantities such an

evaluation can be misleading. As discussed above for the case of boundary layer clouds, parts of the effects of too coarse horizontal resolution on the cloud liquid water amount can be compensated by a too coarse vertical resolution. Similarly, effects of choices in the remaining parameterizations of GSRMs may be larger than effects of vertical resolution changes and obscure potential improvements or degradations of model skills in representing observations. Lang et al. (2023) simulated larger effects from changes in the parameterizations of microphysics and vertical turbulent diffusions on tropospheric temperature

and relative humidity at least in the lower and middle troposphere compared to the effects from our vertical resolution changes in experiments L55, L110, and L190. Nevertheless, future studies should evaluate the effects of vertical resolution on specifics of the representation of different climate features, as it has been done e.g. by Seiki et al. (2015) for tropical cirrus.

A central motivation for the investment in higher horizontal resolution in global atmospheric models is that the step to storm-resolving scales enables to get rid of parameterizations, in particular for convection. Our analysis shows that even when

the storm-resolving scale is reached, for most climate quantities the effect of halving horizontal grid-spacing is larger than the effect of halving vertical grid-spacing, but of the same order of magnitude. As doubling vertical resolution is computationally cheaper than doubling horizontal resolution, we conclude that further computing time investments in vertical refinement may affect truncation errors of GSRMs similarly to comparable investments in horizontal refinement. Furthermore, our study points to the inadequacy of coarser vertical grid spacing than used in our reference grid with 110 layers, and highlights that specific

climate features like boundary layer clouds show no convergence even for a grid with a maximum tropospheric layer thickness of $50\,\mathrm{m}$ which is much finer than what is currently used in most GSRMs.

*Code availability.* Simulations were done with the ICON branch nextgems_cycle1_zstar_avr of the icon-aes repository as commit 2d18086d538ca6b80785f21b7a14808fbf50546c. This source code and run scripts are available here: https://edmond.mpdl.mpg.de/dataset.xhtml?persistentId=doi:10.17617/3.Z10MPA (last access:10 July, 2023). The ICON model is available

to individuals under licenses (https://code.mpimet.mpg.de/projects/iconpublic/wiki/How%20to%20obtain%20the%20model%20code, last access: 10 July, 2023). By downloading the ICON source code, the user accepts the license agreement. Scripts employed to produce the figures can be found here: https://hdl.handle.net/21.11116/0000-000D-605D-F (last access: 04 July 2023).

*Author contributions.* HS and BS developed the idea for this study. The vertical grids were developed and the model code adapted for fine

vertical grid spacing by HS and SR. The simulations were run by SR, HS, and TL. The analysis of the results was conducted by HS, JB, S-WF, DJC, PK, LK, CK, UN, AS, and AW. The paper was written by HS with input, comments, and reviews from all authors.



*Competing interests.* The authors declare that they have no competing interests.

*Acknowledgements.* We thank Ann Kristin Naumann for very helpful comments on an earlier version of this manuscript. We thank René Redler and Karl-Hermann Wieners for useful recommendations for running the simulations. The simulations were run at the German Climate
Computing Center (DKRZ) and we thank the DKRZ staff for their support.



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



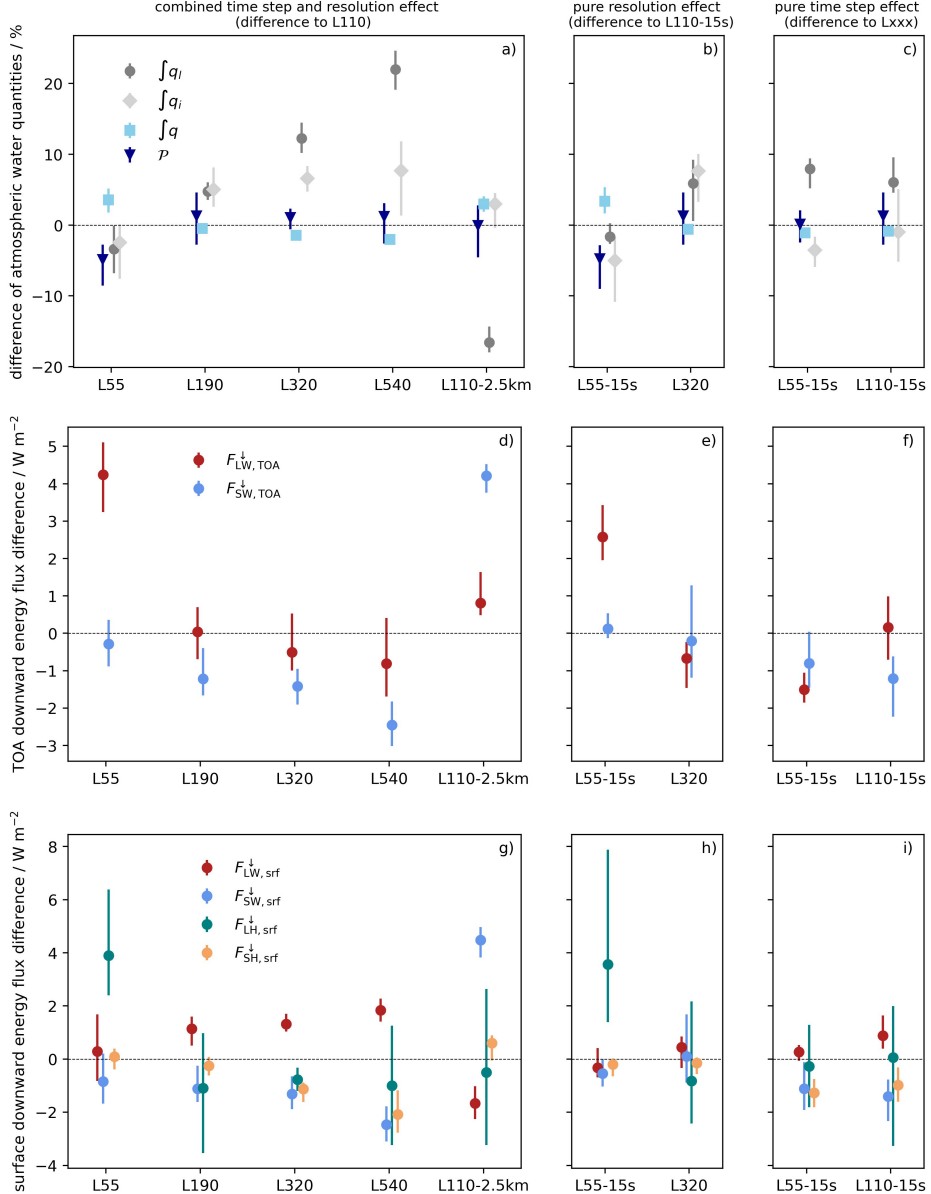

**Figure 2.** Panels a), d), and g) show differences of globally averaged atmospheric quantities from simulations that differ, except for (L55 - L110), in both vertical grid spacing and length of the integration time step. Panels b), e), and h) show effects from differences in vertical grid spacing only, and Panels c), f), and i) effects from differences in the time step, only. Differences are calculated between the model configuration indicated at the x-axis, and L110 (a, d, g), L110-15s (b, e, h), and the configuration with the same vertical grid spacing as the configuration on the x-axis but longer time step (c, f, i), respectively. Quantities depicted are vertically integrated cloud liquid water and cloud ice, precipitable water, and precipitation ($\int q_l$, $\int q_i$, $\int q$, $\mathcal{P}$; a to c), net long-wave, and short-wave radiation at the TOA ($F^{\downarrow}_{\mathrm{LW,TOA}}$, $F^{\downarrow}_{\mathrm{SW,TOA}}$; d to f), and long-wave and short-wave radiation, sensible, and latent heat fluxes ($F^{\downarrow}_{\mathrm{LW,srf}}$, $F^{\downarrow}_{\mathrm{SW,srf}}$, $F^{\downarrow}_{\mathrm{SH,srf}}$, $F^{\downarrow}_{\mathrm{LH,srf}}$; g to i) at the surface. All fluxes are defined positive downward. Markers depict averages over 40 days, the vertical bars mark the minimum and maximum differences of the four 10-day subperiods.




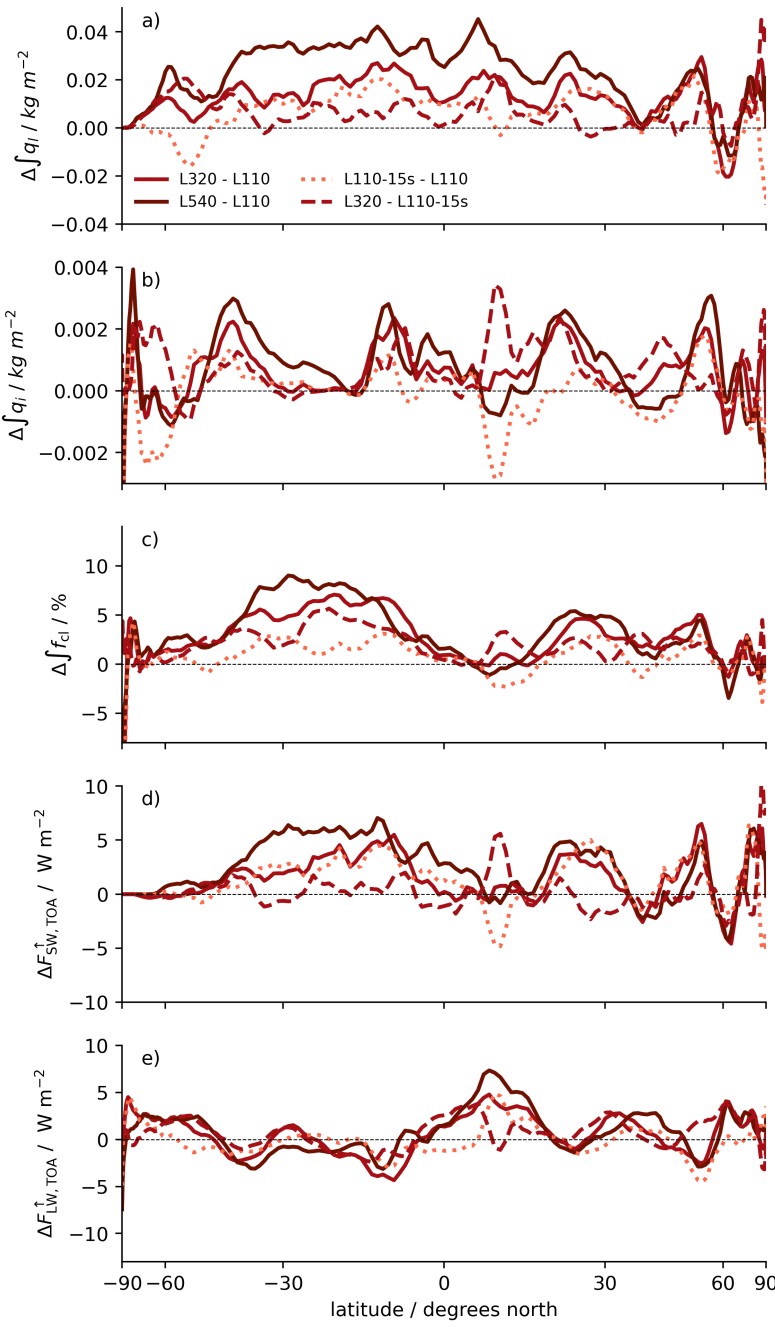

**Figure 3.** Differences of zonally averaged atmospheric quantities between simulations as indicated in the legend of Panel a). The quantities are a) vertically integrated cloud liquid water $\int q_l$, b) vertically integrated cloud ice $\int q_i$, c) total cloud fraction $\int f_{cl}$, d) outgoing shortwave radiation at TOA $F^{\uparrow}_{\mathrm{SW,TOA}}$, and e) outgoing longwave radiation at TOA $F^{\uparrow}_{\mathrm{LW,TOA}}$. Solid lines mark differences of practical vertical resolution changes (i.e. in general also including a time step change), the dashed line a difference caused by a pure vertical resolution change, and the dotted line a difference caused by a pure time step change. All quantities are averaged over the last 40 days of the simulations.





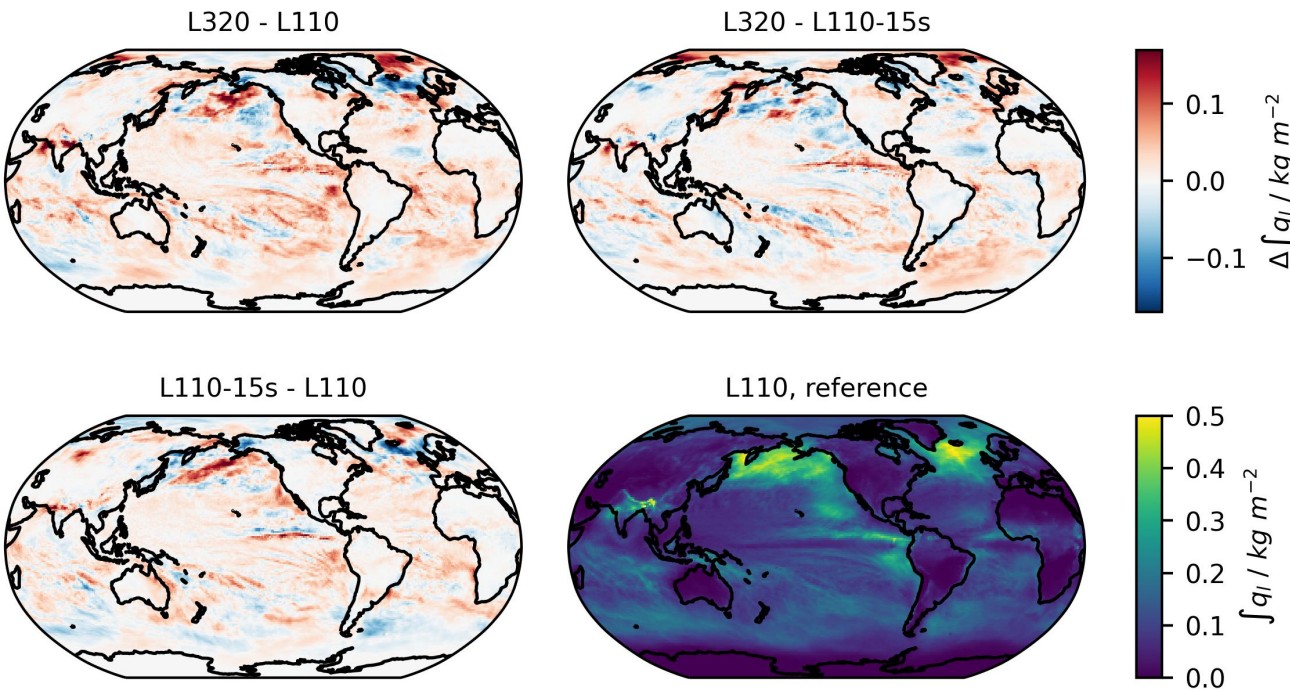

**Figure 4.** Vertically integrated cloud liquid water ($\mathrm{kg\,m^{-2}}$) from simulation L110 (lower right panel) and differences in this quantity between the simulations indicated in the panel titles. All quantities are averaged over the last 40 days of the simulations.



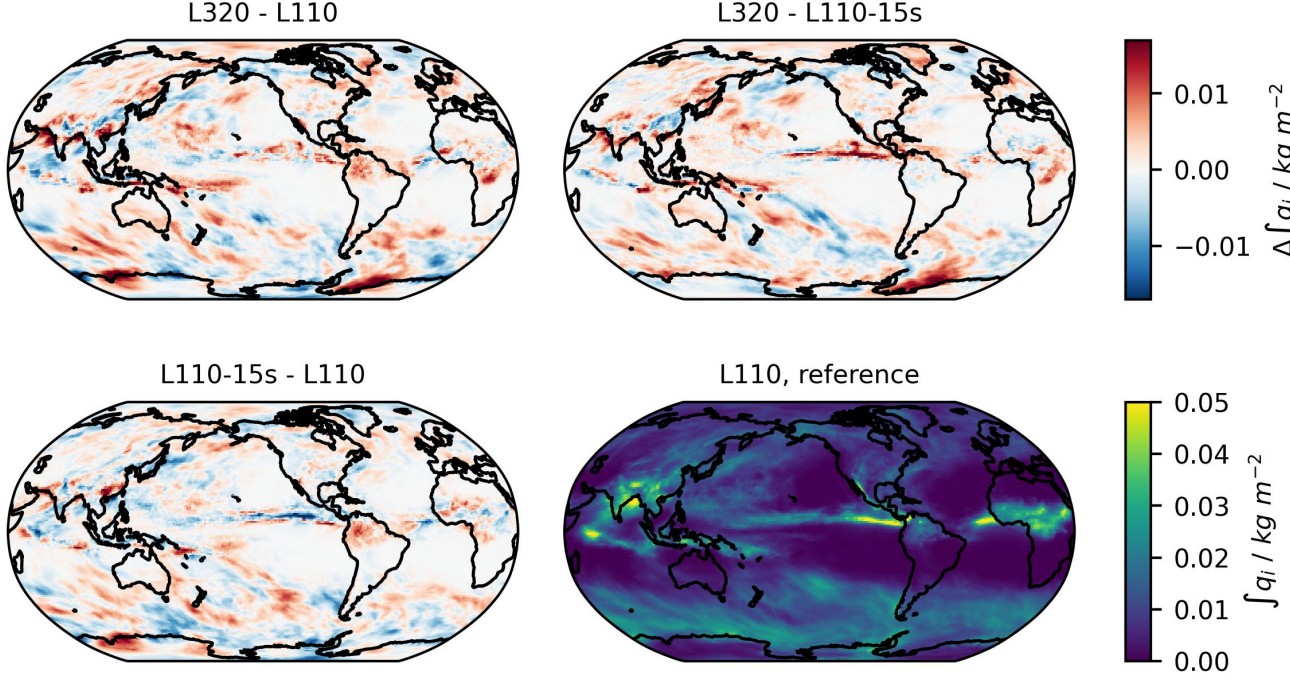

**Figure 5.** Vertically integrated cloud ice $(\mathrm{kg\,m^{-2}})$ from simulation L110 (lower right panel) and differences in this quantity between the simulations indicated in the panel titles. All quantities are averaged over the last 40 days of the simulations.



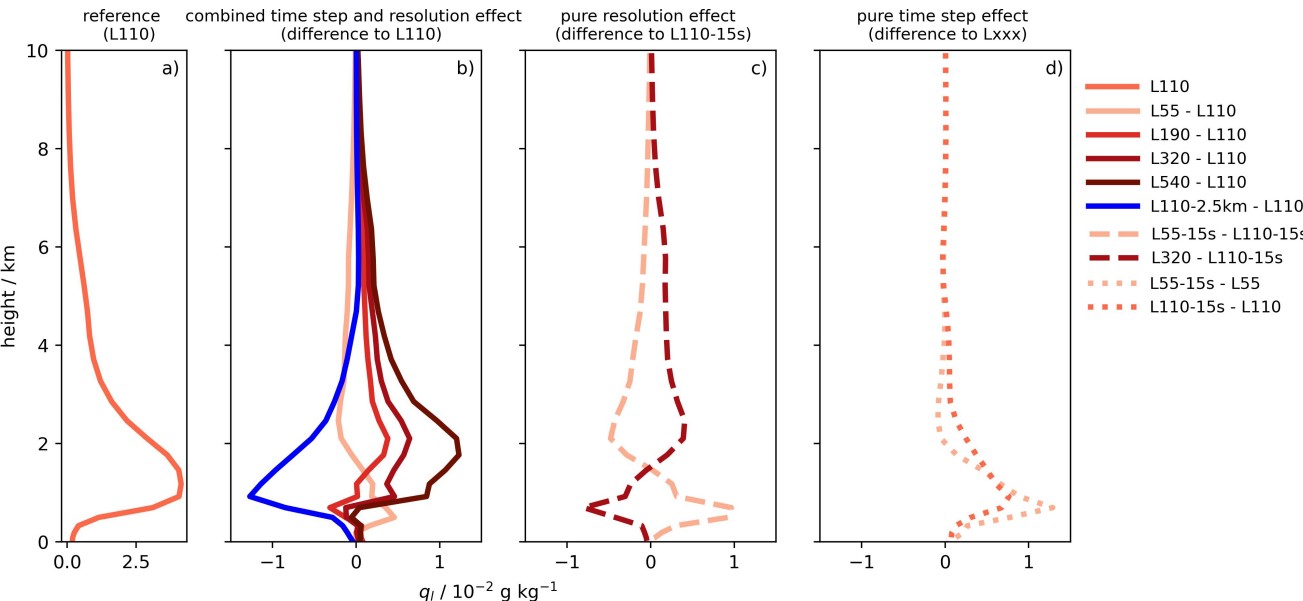

**Figure 6.** Vertical profiles of the cloud liquid water concentration ($10^{-2}\,\mathrm{g\,kg^{-1}}$) averaged over the tropics (30°S to 30°N) from simulation L110 (panel a) and effects on this quantity resulting (panel a) from combined vertical resolution and time step changes, from pure vertical resolution changes (panel c), and from pure time step changes (panel d). Profiles in panel b) are calculated as differences between the simulation indicated in the legend and simulation L110, panel c) shows differences between the simulation indicated in the legend and simulation L110-15s, and panel d) shows the differences (L55-15s - L55) and (L110-15s - L110). To compare the different grids, all values have been interpolated to the L55 grid.





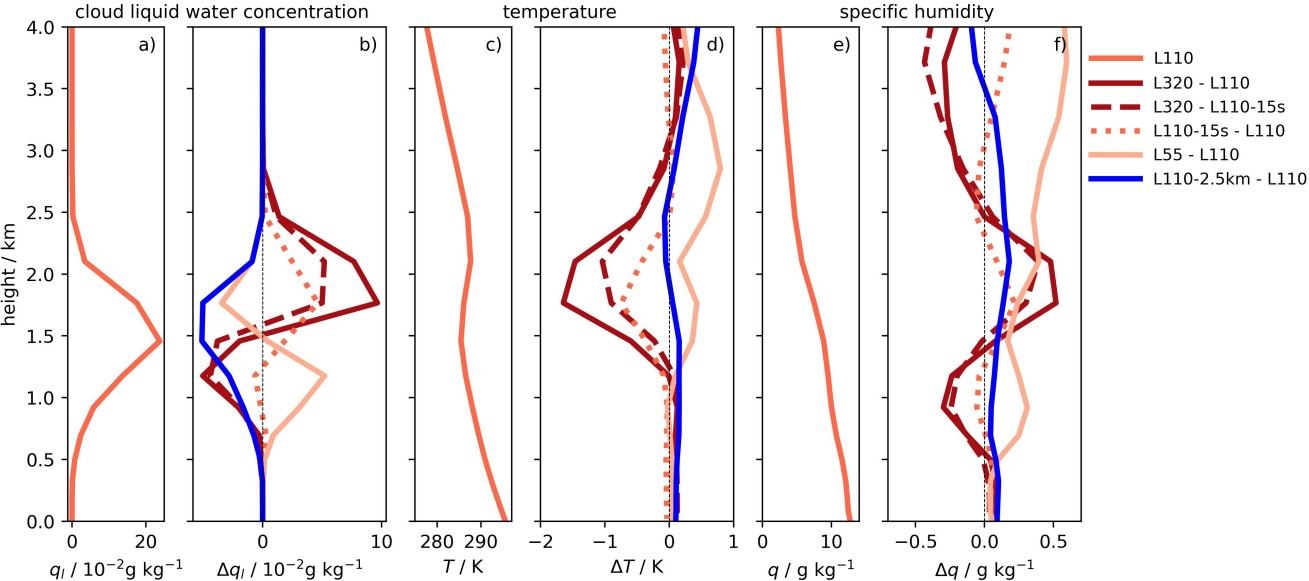

**Figure 7.** Vertical profiles of (a) cloud liquid water concentration ($q_i$; $10^{-2}\,\mathrm{g\,kg^{-1}}$), (c) temperature ($T$; K), and (e) specific humidity ($q$; $\mathrm{g\,kg^{-1}}$) from L110 averaged over the region 15°S to 0°N and 105°W to 90°W, i.e. a stratocumulus region off the South American coast. Panels b), d), and f) show the differences in the respective quantities for a practical increase of the vertical resolution (L320 - L110; solid dark red), a pure increase of vertical resolution (L320 - L110-15s; dashed dark red), a pure time step decrease (L110-15s - L110; dotted red), a decrease of the vertical resolution (L55 - L110; solid light red), and a practical increase of the horizontal resolution (L110-2.5km - L110; solid blue).




**Figure 8.** Color shading indicates the difference of vertical profiles of the cloud liquid water concentration $q_i$ between experiments a) L320 - L110, b) L110-15s - L110, and c) L320 - L110-15s averaged over percentile bins of tropical precipitation. Numbers on the x-axis mark the percentiles: 10, e.g., marks the mean profile for the tropical $1°$x$1°$ areas and simulated days with the 10% lowest precipitation rates, 100 marks the 10% highest precipitation rates. Black solid lines mark average $q_i$ values of 2, 4, and $6 \times 10^{-2}\,\mathrm{g\,kg^{-1}}$ in L110.





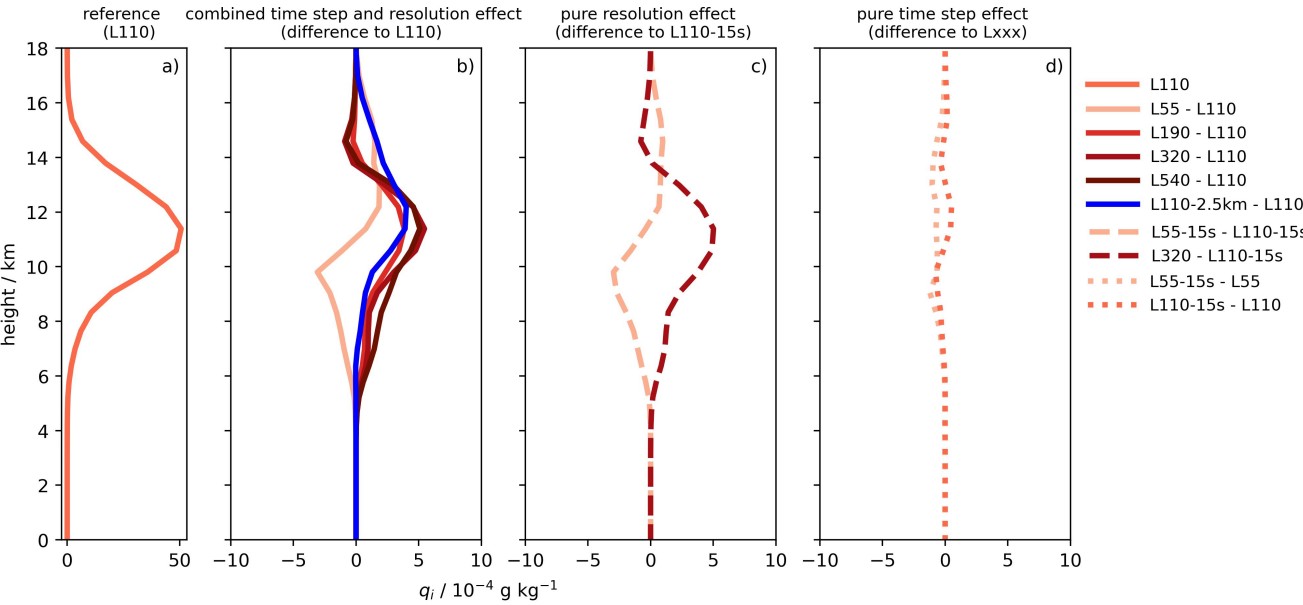

**Figure 9.** As Fig. 6 but for the cloud ice concentration ($10^{-4}\,\mathrm{g\,kg^{-1}}$).



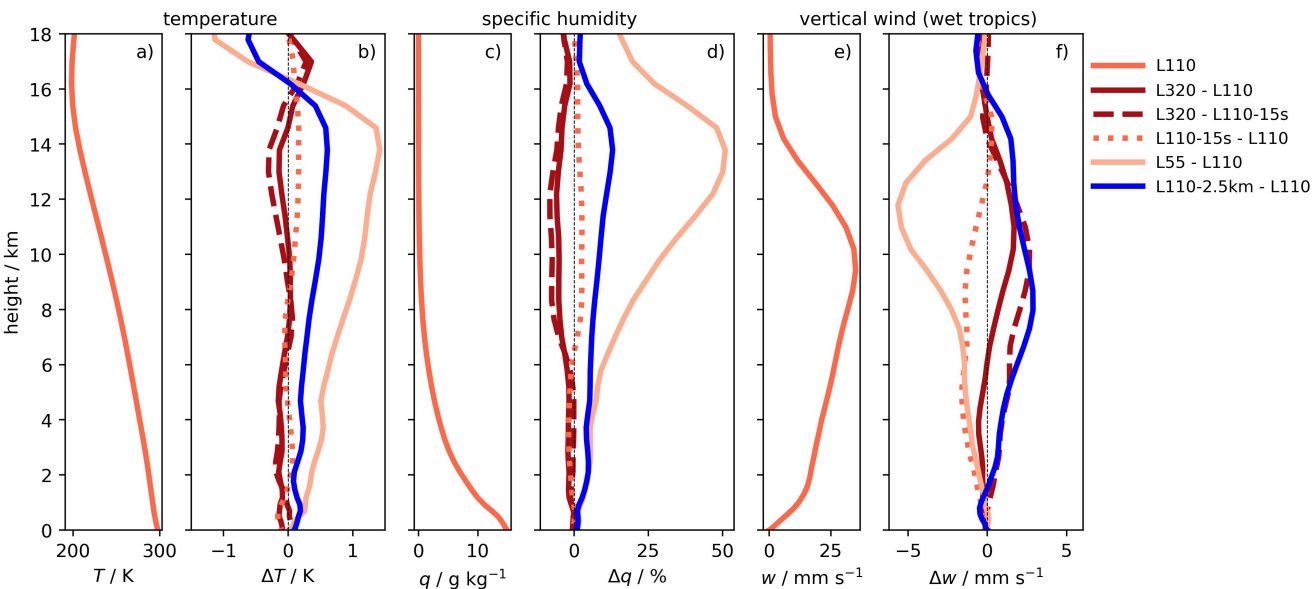

**Figure 10.** Vertical profiles of (a) temperature ($T$; K), (c) specific humidity ($q$; g kg$^{-1}$), and (e) vertical wind ($w$; m s$^{-1}$) from L110 averaged over the tropics (30°S to 30°N), for the vertical wind data is averaged over the 10% of the tropical area with the highest precipitation. Panels b), d), and f) show the differences in the respective quantities for a practical increase of the vertical resolution (L320 - L110; solid dark red), a pure increase of vertical resolution (L320 - L110-15s; dashed dark red), a pure time step decrease (L110-15s - L110; dotted red), a decrease of the vertical resolution (L55 - L110; solid light red), and a practical increase of the horizontal resolution (L110-2.5km - L110; solid blue).





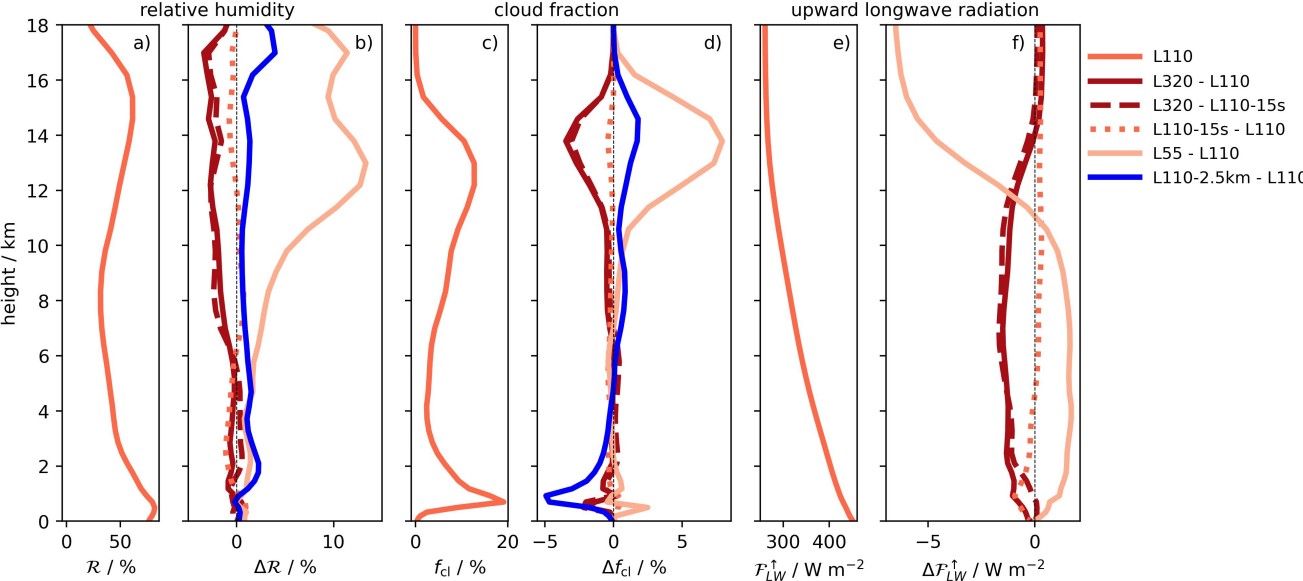

**Figure 11.** As Fig. 10 but for the tropical averages of (a) relative humidity ($\mathcal{R}$; %), (c) cloud fraction ($f_{\mathrm{cl}}$; %), and (e) upward longwave radiation flux ($\mathcal{F}_{\mathrm{LW}}^{\uparrow}$; W m$^{-2}$). Differences for relative humidity and cloud fraction are given in percentage points. Due to limited model output radiation fluxes are averaged over 30 days only (instead of 40 days used for all other time averages) and not given for L110-2.5km.



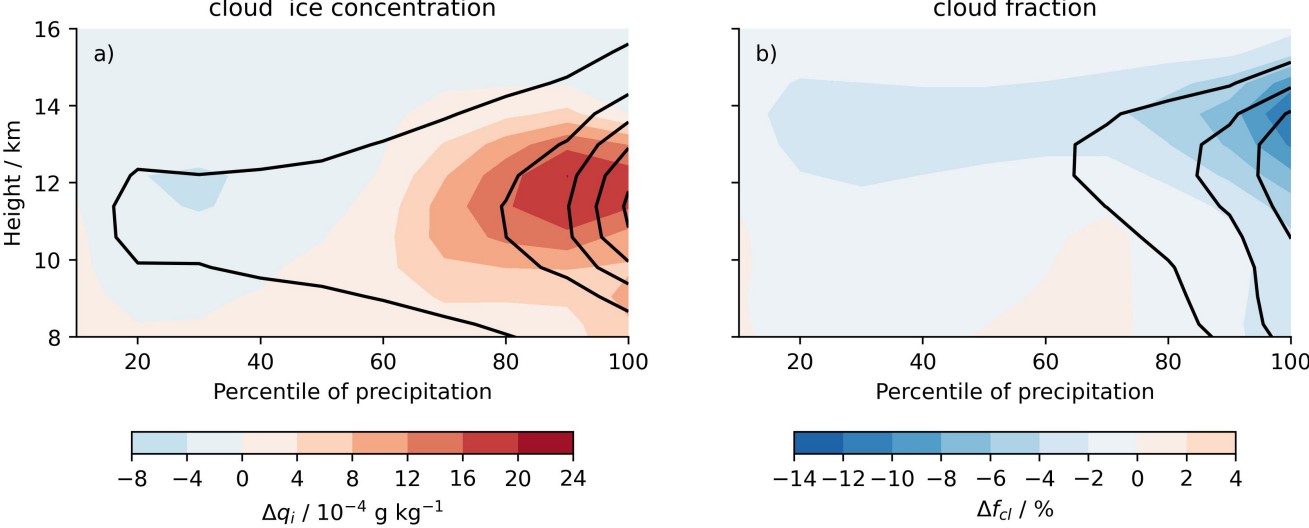

**Figure 12.** Left: Color shading indicates the difference of vertical profiles of the cloud ice concentration $q_i$ between experiments L540 and L110 averaged over percentile bins of tropical precipitation. Numbers on the x-axis mark the percentiles: 10, e.g., marks the mean profile for the tropical $1°\text{x}1°$ areas and simulated days with the 10% lowest precipitation rates, 100 marks the 10% highest precipitation rates. Black solid lines mark average $q_i$ values of 10, 60, 110, 160, and $210 \times 10^{-6}\,\text{g kg}^{-1}$ in L110. Right: As left, but for the cloud fraction $f_{cl}$ in %. Black isolines mark average $f_{cl}$ values of 10, 20, and 30%.

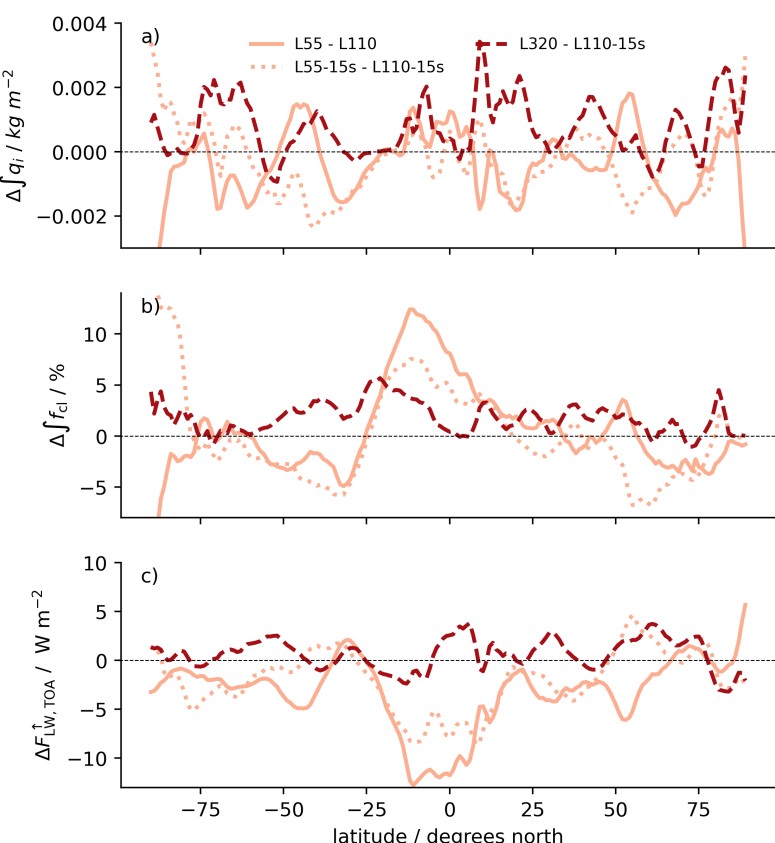

**Figure 13.** Differences of zonally averaged atmospheric quantities between simulations as indicated in the legend of Panel a). The quantities are a) vertically integrated cloud ice $\int q_i$, b) total cloud fraction $\int f_{\mathrm{cl}}$, and c) outgoing longwave radiation at the TOA $F^{\downarrow}_{\mathrm{LW,TOA}}$. Solid lines mark differences of practical vertical resolution changes (i.e. in general also including a time step change), the dashed line a difference caused by a pure vertical resolution change, and the dotted line a difference caused by a pure time step change. All quantities are averaged over the last 40 days of the simulations.