# Peer review of "Effects of vertical grid spacing on the climate simulated in the ICON-Sapphire global storm-resolving model"

_EGUsphere, 2023_

## Author Comment (AC1)

**Response to reviewers**

November 24, 2023

We thank the three reviewers for their thorough and very constructive reviews of our manuscript "Effects of vertical grid spacing on the climate simulated in the ICON-Sapphire global storm-resolving model". Below we cite the original reviews in blue and add our replies in black font.

**1 Reviewer 1**

**1.1 General comments**

This study describes the vertical resolution dependency of the global storm-resolving simulation for the various vertical grid spacings between 800 m and 50 m. Because the time step is generally chosen for the vertical resolution, the authors further examined the time step dependency and the vertical resolution dependency with the fixed time step. The comparison for the vertical and horizontal resolution dependencies is added. Among the comprehensive results described, both the cloud liquid water and the cloud ice amount increase with the vertical resolution, and these do not converge until the 50 m vertical grid spacing. Cloud liquid water is equally dependent on the vertical resolution and the time step. The effect on the radiative fluxes is also documented, and the possible reason and the mechanism are argued.

This paper stands out for conducting numerous experiments across various vertical resolutions and effectively delving into the dependency on time steps and vertical resolutions as distinct discussion points. The analysis of mechanisms concerning the results is also reasonable and seems well-justified. With minor revisions, I believe this paper is ready for publication.

**1.2 Specific comments**

p. 2, L.23: Clarify what kind of "the effect of a halving of the grid spacing".

The following sentence of the original manuscript provides information on two "effects" for the halving of grid spacing from 5 to 2.5 km. We will add information on halving at larger grid spacings so that it becomes clear that this sentence provides examples for "the effect".

p. 9, L.243: Please insert references for "past studies".

We will add references to Marchand and Ackermann (2011) and Bogenschutz et al. (2021).

p. 9, L. 248: It is unclear where a strong stratocumulus signal contributes in "the South Pacific" in Fig. 4.

We agree. This was a formulation error that we will remove.

p. 10, L.260-267: Please quantify the correlation between cloud liquid water and reflected shortwave radiation and the other correlations. The correlation may be defined at each latitude, and the latitudinal profile of the correlation can be shown. Interestingly, reflected shortwave radiation correlates well with cloud ice in some latitudes, and OLR correlates more with the cloud fraction.

We will add a new panel to the manuscript's Figure 3 that includes correlation coefficients of reflected SW radiation with vertically integrated cloud ice, cloud liquid water, and cloud fraction, see Fig. 1 of this reply. We think that this corroborates the claims we made in our original manuscript, in particular the dip in correlation of cloud liquid water and reflected SW near the ITCZ. We agree that the partly differing correlations of OLR with cloud ice on the one hand, and with cloud fraction on the other hand are also interesting. However, we would restrict the quantification of correlations here to the SW radiation because the other point is already discussed in Section 3.3.2.

p. 11, L. 284: It should be clearer to rewrite as "the effect of a spatial refinement on cloud liquid water".

We will change the sentence to "effect of a spatial refinement on cloud condensate" because the reasoning could be applied to both the liquid and frozen components.

p. 11, L. 315-381: In the Northern Pacific, the cloud liquid water effects vary regionally. The reviewer speculates that this regional variation is related to the synoptic condition simulated in this period. What do the authors mean by "the ventilation of the boundary layer"? It is not convincing that the numerical diffusion and "the ventilation of the boundary layer" explain the regional variation of the difference of cloud liquid water.

The point we wanted to make here is that an increase of cloud liquid water with increased vertical resolution occurs over large parts of the globe under very different conditions. Similar mechanisms may be at play under these different conditions, among them the "ventilation" of the boundary layer. However, indeed, our formulation obscured this point. we will try to rephrase this paragraph and mention the "ventilation" argument earlier when referring to past studies in 3.2.

p. 12, L. 338-346: Figure 11f is not referred to in the text. Why the increase of upwelling (Fig. 10f) is related to the drying of the troposphere? It cannot be seen where atmospheric radiative cooling increases from Fig. 11f.

Figure 11f is actually referred to, but only later in the manuscript, original line 404. The purpose of showing 11f is to support the idea that the change in OLR is not related to the ice cloud content but the ice cloud fraction. It's true that the radiative cooling can't be inferred from Fig. 11f. We will add the information "not shown" to the radiative cooling in order not to confuse the reader with Fig. 11f.

**2 Reviewer 2**

**2.1 General comments**

The study by Schmidt et al. performed a series of 45-day simulations using the global storm-resolving model ICON-Sapphire and analyzed the sensitivities of some basic "climate" features to the choices of vertical grid spacing and time step length. The manuscript documents sensitivities in the simulated atmospheric water amount, energy fluxes, air temperature, etc. The quantities presented include global integrals, zonal averages, geographical distributions, as well as mean vertical profiles in selected regions and different precipitation regimes.

I applaud the authors' interest in carrying out such a study, and I appreciate the large amount of resources, including both computing time and human hours, invested in the study. The manuscript is relatively easy to follow, and the results can serve as a useful reference for other models of this kind. Therefore, my overall recommendation is to publish the manuscript in GMD after a round of minor revision.

The authors documented resolution sensitivities in this manuscript but did not provide in-depth explanations of the causes of such sensitivities. This is understandable given the complexity of the numerical model. Furthermore, while ICON-Sapphire is a global storm-resolving model, the manuscript focused heavily on spatially and temporally aggregated results without showing sensitivities in the simulated storm characteristics. It would be very useful if the authors could carry out some future studies in those directions.

We agree and are currently thinking about further studies.

**2.2 Specific comments**

Because of the many simulations discussed in the manuscript and the multiple ways of comparison performed (i.e., time and space combined and in isolation), it would be helpful to always include the time step length in the simulation names. I.e., in Table 1, in all figures except the first one, and throughout the text, it would be useful to say "L110-40s" and "L320-15s" instead of simply "L110" and "L320".

The original thinking was that we would use only the number of layers to characterize the experiments we describe as "practical" resolution change experiments. However, we understand that it may be more convenient to add the time step length and will do that in the revised manuscript.

By eyeballing Figure 1, we can easily discern the two or three sectors that constitute each vertical grid. As the number of grid layers increases, we see more and more examples of discontinuous slopes along the curves. I'm curious whether these discontinuities have been found to (or are expected to) affect the vertical propagation of waves. Could these discontinuities have contributed to the numerical instabilities in L320 and L540 simulations mentioned at line 133 of the preprint? If the authors had a chance to redesign and rerun the simulations, would they construct the grids differently?

No, we wouldn't construct them very differently. Within the limits of construction rules for the ICOn grid we still think our choice is sensible for the purpose of the study. We don't think that the apparent discontinuities affect the stability. We have constructed the grids in a way that thickness changes from one layer to another are relatively small. In the search for instabilities mentioned in the manuscript we had tested also grids without the apparent discontinuities but this didn't solve the problem. We will add a respective remark to the manuscript.

*In Figure 1, the curves are of similar colors and many segments are on top each other. It would be useful to use more colors - and reduce the marker sizes - to improve the visibility of all grid configurations.*

We have tested several options and finally only reduced the size of the cross markers (see Fig. 2). This makes overlaps easier to identify. However, we would like to keep the colors to use consistent coloring for the same vertical grid throughout the manuscript. We'd also like not to use different colors but keep the red hues. Different colors would indeed be easier to identify but it would be much less intuitive to compare different grid spacing.

*The abstract only summarizes the numerical results. It would be useful to include some of the conclusions from the last section, particularly on the choices of grid configuration for future simulations.*

We will add the following sentences, condensed from the final paragraph of the manuscript, to the abstract:

"The simulations show that using larger maximum tropospheric vertical grid spacing than 400 m would increase the truncation error strongly. Computing time investments in a further vertical refinement may affect truncation errors of GSRMs similarly to comparable investments in horizontal refinement, because halving vertical grid spacing is in general cheaper than halving horizontal grid spacing. However, convergence of boundary layer cloud properties can't be expected even for the the smallest maximum tropospheric grid spacing of 50 m used in this study."

**3  Reviewer 3**

**3.1  General comment**

*This paper focuses on the effects of vertical resolution on climate simulations using the ICON-Sapphire global storm-resolving model (GSRM). The authors find that while the model has sensitivity to vertical resolution (and time step), it is not as large as the sensitivity of changing the horizontal resolution. I commend the authors for all the hard work put into performing and analyzing these simulations. The topic of vertical resolution is often forgotten about when compared to horizontal resolution sensitivity and this is the first robust study (that I know of) for GSRMs. Ultimately, I do feel this paper should be published, but I hope the authors consider my recommendations that I feel will increase the impact and clarity of this work.*

**3.2  Major comments**

*I strongly encourage the authors to provide analysis and discussion regarding the performance of their vertical grid configurations when compared to observations. I realize the authors admit in the conclusions they choose not to do this, but it makes the paper feel incomplete in my opinion. Having no observational reference makes these experiments feel somewhat arbitrary and makes the paper feel a little boring in spots. I do not feel this needs to dominate the paper, but having some analysis to answer the question of "does higher vertical resolution lead to better results (and by how much)?" would help to increase the impact of this paper and make it more interesting to other modeling centers.*

This comment doesn't come unexpected. We had actually discussed this earlier among the authors of the paper, but had decided to not show any comparison mainly for the reasons discussed in the manuscript, but also we thought the manuscript is more than long enough without it. However, motivated by the reviewer's comment, we decided to add some comparison in a new section of the paper. We will add Fig. 3 of this reply to the manuscript. The figure actually confirms the issue of a possible compensation of errors. Clearly, the temperature profile of the L55 simulation, i.e. the simulation with the coarsest grid, is closest to radiosonde observations. This is not the case for specific humidity which is in general overestimated by L55 and underestimated by the other simulations. We will also add a comparison to CERES TOA energy fluxes to the revised manuscript.

*As the paper is written, it feels more like a sensitivity paper specific only to the Sapphire model. As a model developer myself, I wished the authors would have done a better of job of discussing*

and hypothesizing how generalizable their results may be for other GSRMs. I realize this is a rather difficult thing to do but some suggestions to bring this to fruition could be to 1) discuss if Sapphire's parameterizations have been tested offline to satisfy vertical resolution convergence. Whether they have or have not would be an interesting data point to other modeling centers about the type of sensitivity they may expect from their models if they followed similar testing approaches (or not). 2) Discuss if any parameterizations in Sapphire have explicit dependence on the vertical grid spacing (aside from discretization, of course). Example: the SAM cloud resolving model (Khairoutdinov and Randall 2003) caps the SGS turbulence length scale to the local vertical grid spacing. This inevitably leads to a vertical grid sensitivity. If Sapphire has something like this (I see it uses a similar turbulence scheme to SAM but couldn't find a paper with the specifics), this would be worthwhile to document and could explain some sensitivity seen in this paper. 3) The authors could speculate on the choice of SGS schemes to vertical resolution sensitivity (i.e. would a more complex turbulence scheme perhaps lead to a more robust solution?).

We would like the paper to be useful for other readers than just ICON users. This is why we tried to point to similarities and differences with earlier studies to provide an idea how robust or model-dependent specific results may be. However, conclusions on robustness can only be made very cautiously because model configurations and experimental setups differ. We will follow the suggestion of the reviewer and provide more information on the turbulent mixing scheme, which indeed contains a mixing length estimation that depends on horizontal and vertical grid spacing. ICON parameterizations have not been tested offline for dependencies on grid spacings but we agree that this would be very useful to better understand the sensitivities simulated in our experiments and will add a respective comment in the final section.

While I found the paper to be well written enough that I could understand what the authors were trying to say, I did find many paragraphs and passages that were very awkwardly worded that required me to read them several times before I got the point. One (of many) examples of this is page 11, paragraph 305. Therefore, I encourage the authors to read through the document and improve the wording and English.

We will do this before submitting the revised version.

**3.3 Minor comments**

This paper finds that the South America stratocumulus are the most responsive to vertical resolution change. This is interestingly very consistent with the findings of Bogenschutz et al. (2021) and Lee et al. (2021; 2022) in E3SM and I feel this should be explicitly noted.

We will add such a remark and references to these manuscripts.

Further, the above mentioned work notes that significant improvement to marine Sc (in their models) is not achieved until the vertical grid spacing approaches 15 m in the lower troposphere. Since observational references are not provided in this paper, it's hard to tell if Sapphire's increased vertical grid (which is not as aggressive in the lower troposphere compared to the aforementioned works) significantly improves marine Sc or not. Once observational references are added, it may be interesting to tie the findings with that of the E3SM work.

We already referred to the findings of Bogenschutz et al. (2021) and Lee et al. (2022) in several places of the manuscript. However, we agree that the explicit information on the vertical grid spacing of 15 m which is smaller than our smallest grid spacing is useful and we will add this information.

The authors may want to consider citing the work of Cheng et al. (2010) https://doi.org/10.3894/JAMES.2010.2.3 which is a nice paper that lays the groundwork for vertical and horizontal resolution sensitivity for CRMs in the doubly periodic framework.

We agree that a reference to this paper is useful. It will be added.

[Figure]

Figure 1: Panels a) to e) show differences of zonally averaged atmospheric quantities between simulations as indicated in the legend of Panel a). The quantities are a) vertically integrated cloud liquid water $\int q_l$, b) vertically integrated cloud ice $\int q_i$, c) total cloud fraction $\int f_{cl}$, d) outgoing shortwave radiation at TOA $F^{\uparrow}_{SW,TOA}$, and e) outgoing longwave radiation at TOA $F^{\uparrow}_{LW,TOA}$. Solid lines mark differences of practical vertical resolution changes (i.e. in general also including a time step change), the dashed line a difference caused by a pure vertical resolution change, and the dotted line a difference caused by a pure time step change. All quantities are averaged over the last 40 days of the simulations. Panel f) shows coefficients of correlations between $F^{\uparrow}_{SW,TOA}$ on the one, and $\int q_l$, $\int q_i$, and $\int f_{cl}$ on the other hand, calculated for each latitude over all experiments of Table 1.

[Figure]

Figure 2: Vertical layer thickness for the examples of grid boxes with surface altitudes of 0 (circles) and 8205 (crosses), respectively. The y-axis shows the height of the lower edge of the respective layer. Colors indicate the experiments (see Table **??** for details) as indicated in the legend. Darker colors mark finer vertical grid spacing. Only the lowest 25 of the grids are shown. The model top is at 75 for all configurations.

[Figure]

Figure 3: Mean vertical profiles of temperature (a) and specific humidity (c) from 257 radio soundings taken between June, 29, and August, 6, 2021 in the tropical Atlantic. Panels b) and d) show differences to these observations from mean simulated profiles from the simulations indicated in the legend and sampled at the time and location of the observations.

---

## Author Response (AR1)

**Response to reviewers**

December 21, 2023

We thank the three reviewers for their thorough and very constructive reviews of our manuscript "Effects of vertical grid spacing on the climate simulated in the ICON-Sapphire global storm-resolving model". Below we cite the original reviews in blue and add our replies in black font.

The major change we have made to the original manuscript is arguably the addition of a new Section 6 on the comparison to observations that was suggested by Reviewer 3. This has also led us to include a further co-author, Amrit Cassim, who worked on the model-observation comparison. However, we have tried to improve the manuscript also with respect to all other issues raised by the reviewers.

**1 Reviewer 1**

**1.1 General comments**

This study describes the vertical resolution dependency of the global storm-resolving simulation for the various vertical grid spacings between 800 m and 50 m. Because the time step is generally chosen for the vertical resolution, the authors further examined the time step dependency and the vertical resolution dependency with the fixed time step. The comparison for the vertical and horizontal resolution dependencies is added. Among the comprehensive results described, both the cloud liquid water and the cloud ice amount increase with the vertical resolution, and these do not converge until the 50 m vertical grid spacing. Cloud liquid water is equally dependent on the vertical resolution and the time step. The effect on the radiative fluxes is also documented, and the possible reason and the mechanism are argued.

This paper stands out for conducting numerous experiments across various vertical resolutions and effectively delving into the dependency on time steps and vertical resolutions as distinct discussion points. The analysis of mechanisms concerning the results is also reasonable and seems well-justified. With minor revisions, I believe this paper is ready for publication.

**1.2 Specific comments**

p. 2, L.23: Clarify what kind of "the effect of a halving of the grid spacing".

The subsequent sentence of the original manuscript provides information on two "effects" for the halving of grid spacing from 5 to 2.5 km. We've added information on halving at larger grid spacings so that it becomes clear that this sentence provides examples for "the effect".

p. 9, L.243: Please insert references for "past studies".

We've added references to Marchand and Ackermann (2011) and Bogenschutz et al. (2021).

p. 9, L. 248: It is unclear where a strong stratocumulus signal contributes in "the South Pacific" in Fig. 4.

We agree. This was a formulation error that we've removed.

p. 10, L.260-267: Please quantify the correlation between cloud liquid water and reflected shortwave radiation and the other correlations. The correlation may be defined at each latitude, and the latitudinal profile of the correlation can be shown. Interestingly, reflected shortwave radiation correlates well with cloud ice in some latitudes, and OLR correlates more with the cloud fraction.

We have added a new panel to the manuscript's Figure 3 that includes correlation coefficients of reflected SW radiation with vertically integrated cloud ice, cloud liquid water, and cloud fraction, see Fig. 1 of this reply. We think that this corroborates the claims we made in our original manuscript, in particular the dip in correlation of cloud liquid water and reflected SW near the ITCZ. We agree that the partly differing correlations of OLR with cloud ice on the one hand, and with cloud fraction on the other hand are also interesting. However, we restrict the quantification of correlations here to the SW radiation because the other point is already discussed in Section 3.3.2.

 It should be clearer to rewrite as "the effect of a spatial refinement on cloud liquid water".

We've changed the sentence to "effect of a spatial refinement on cloud condensate" because the reasoning could be applied to both the liquid and frozen components.

 In the Northern Pacific, the cloud liquid water effects vary regionally. The reviewer speculates that this regional variation is related to the synoptic condition simulated in this period. What do the authors mean by "the ventilation of the boundary layer"? It is not convincing that the numerical diffusion and "the ventilation of the boundary layer" explain the regional variation of the difference of cloud liquid water.

The point we wanted to make here is that an increase of cloud liquid water with increased vertical resolution occurs over large parts of the globe under very different conditions. Similar mechanisms may be at play under these different conditions, among them the "ventilation" of the boundary layer. However, indeed, our formulation obscured this point. We've rephrased this paragraph and now mention the "ventilation" argument earlier when referring to past studies in 3.2.

 Figure 11f is not referred to in the text. Why the increase of upwelling (Fig. 10f) is related to the drying of the troposphere? It cannot be seen where atmospheric radiative cooling increases from Fig. 11f.

Figure 11f is actually referred to, but only later in the manuscript, original line 404. The purpose of showing 11f is to support the idea that the change in OLR is not related to the ice cloud content but the ice cloud fraction. It's true that the radiative cooling can't be inferred from Fig. 11f. We've added the information "not shown" to the radiative cooling in order not to confuse the reader with Fig. 11f.

**2    Reviewer 2**

**2.1    General comments**

The study by Schmidt et al. performed a series of 45-day simulations using the global storm-resolving model ICON-Sapphire and analyzed the sensitivities of some basic "climate" features to the choices of vertical grid spacing and time step length. The manuscript documents sensitivities in the simulated atmospheric water amount, energy fluxes, air temperature, etc. The quantities presented include global integrals, zonal averages, geographical distributions, as well as mean vertical profiles in selected regions and different precipitation regimes.

I applaud the authors' interest in carrying out such a study, and I appreciate the large amount of resources, including both computing time and human hours, invested in the study. The manuscript is relatively easy to follow, and the results can serve as a useful reference for other models of this kind. Therefore, my overall recommendation is to publish the manuscript in GMD after a round of minor revision.

The authors documented resolution sensitivities in this manuscript but did not provide in-depth explanations of the causes of such sensitivities. This is understandable given the complexity of the numerical model. Furthermore, while ICON-Sapphire is a global storm-resolving model, the manuscript focused heavily on spatially and temporally aggregated results without showing sensitivities in the simulated storm characteristics. It would be very useful if the authors could carry out some future studies in those directions.

We agree and are currently thinking about further studies.

**2.2    Specific comments**

Because of the many simulations discussed in the manuscript and the multiple ways of comparison performed (i.e., time and space combined and in isolation), it would be helpful to always include the time step length in the simulation names. I.e., in Table 1, in all figures except the first one, and throughout the text, it would be useful to say "L110-40s" and "L320-15s" instead of simply "L110" and "L320".

The original thinking was that we would use only the number of layers to characterize the experiments we describe as "practical" resolution change experiments. However, we understand that it may be more convenient to include also information on the time step length and have applied these changes everywhere in the manuscript.

By eyeballing Figure 1, we can easily discern the two or three sectors that constitute each vertical grid. As the number of grid layers increases, we see more and more examples of discontinuous slopes along the curves. I'm curious whether these discontinuities have been found to (or are

No, we wouldn't construct them very differently. Within the limits of construction rules for the ICON grid we still think our choice is sensible for the purpose of the study. We don't think that the apparent discontinuities affect the stability. We have constructed the grids in a way that thickness changes from one layer to another are relatively small. In the search for instabilities mentioned in the manuscript we had tested also grids without the apparent discontinuities but this didn't solve the problem. We've added a respective remark to the manuscript.

We have tested several options and finally only reduced the size of the cross markers (see Fig. 2). This makes overlaps easier to identify. However, we would like to keep the colors to use consistent coloring for the same vertical grid throughout the manuscript. We'd also like not to use different colors but keep the red hues. Different colors would indeed be easier to identify but it would be much less intuitive to compare different grid spacings.

We've add the following sentences, condensed from the final paragraph of the manuscript, to the abstract:
"The simulations show that using larger maximum tropospheric vertical grid spacing than 400 m would increase the truncation error strongly. Computing time investments in a further vertical refinement may affect truncation errors of GSRMs similarly to comparable investments in horizontal refinement, because halving vertical grid spacing is in general cheaper than halving horizontal grid spacing. However, convergence of boundary layer cloud properties can't be expected even for the the smallest maximum tropospheric grid spacing of 50 m used in this study."

**3 Reviewer 3**

**3.1 General comment**

This paper focuses on the effects of vertical resolution on climate simulations using the ICON-Sapphire global storm-resolving model (GSRM). The authors find that while the model has sensitivity to vertical resolution (and time step), it is not as large as the sensitivity of changing the horizontal resolution. I commend the authors for all the hard work put into performing and analyzing these simulations. The topic of vertical resolution is often forgotten about when compared to horizontal resolution sensitivity and this is the first robust study (that I know of) for GSRMs. Ultimately, I do feel this paper should be published, but I hope the authors consider my recommendations that I feel will increase the impact and clarity of this work.

**3.2 Major comments**

I strongly encourage the authors to provide analysis and discussion regarding the performance of their vertical grid configurations when compared to observations. I realize the authors admit in the conclusions they choose not to do this, but it makes the paper feel incomplete in my opinion. Having no observational reference makes these experiments feel somewhat arbitrary and makes the paper feel a little boring in spots. I do not feel this needs to dominate the paper, but having some analysis to answer the question of "does higher vertical resolution lead to better results (and by how much)?" would help to increase the impact of this paper and make it more interesting to other modeling centers.

This comment doesn't come unexpected. We had actually discussed this earlier among the authors of the paper, but had decided to not show any comparison mainly for the reasons discussed in the manuscript, but also we thought the manuscript is more than long enough without it. However, motivated by the reviewer's comment, we decided to add some comparison in the new Section 6 of the manuscript. We include Figs. 3 and 4 of this reply in the manuscript. The figures actually confirm the issue of a possible compensation of errors. Fig. 3,e.g., shows clearly that the temperature profile of the L55 simulation, i.e. the simulation with the coarsest grid, is

closest to radiosonde observations. This is not the case for specific humidity which is in general overestimated by L55=40s and underestimated by the other simulations.

As the paper is written, it feels more like a sensitivity paper specific only to the Sapphire model. As a model developer myself, I wished the authors would have done a better of job of discussing and hypothesizing how generalizable their results may be for other GSRMs. I realize this is a rather difficult thing to do but some suggestions to bring this to fruition could be to 1) discuss if Sapphire's parameterizations have been tested offline to satisfy vertical resolution convergence. Whether they have or have not would be an interesting data point to other modeling centers about the type of sensitivity they may expect from their models if they followed similar testing approaches (or not). 2) Discuss if any parameterizations in Sapphire have explicit dependence on the vertical grid spacing (aside from discretization, of course). Example: the SAM cloud resolving model (Khairoutdinov and Randall 2003) caps the SGS turbulence length scale to the local vertical grid spacing. This inevitably leads to a vertical grid sensitivity. If Sapphire has something like this (I see it uses a similar turbulence scheme to SAM but couldn't find a paper with the specifics), this would be worthwhile to document and could explain some sensitivity seen in this paper. 3) The authors could speculate on the choice of SGS schemes to vertical resolution sensitivity (i.e. would a more complex turbulence scheme perhaps lead to a more robust solution?).

We would very much like the paper to be useful for other readers than just ICON users. This is why we tried to point to similarities and differences with earlier studies to provide an idea how robust or model-dependent specific results may be. However, conclusions on robustness can only be made very cautiously because model configurations and experimental setups differ. We will follow the suggestion of the reviewer and provide more information on the turbulent mixing scheme, which indeed contains a mixing length estimation that depends on horizontal and vertical grid spacing. ICON parameterizations have not been tested offline for dependencies on grid spacings but we agree that this would be very useful to better understand the sensitivities simulated in our experiments. Se have added a respective comment in the of section 2.

While I found the paper to be well written enough that I could understand what the authors were trying to say, I did find many paragraphs and passages that were very awkwardly worded that required me to read them several times before I got the point. One (of many) examples of this is page 11, paragraph 305. Therefore, I encourage the authors to read through the document and improve the wording and English.

We've tried to smoothen the language of the manuscript.

**3.3 Minor comments**

This paper finds that the South America stratocumulus are the most responsive to vertical resolution change. This is interestingly very consistent with the findings of Bogenschutz et al. (2021) and Lee et al. (2021; 2022) in E3SM and I feel this should be explicitly noted.

We've add such a remark to section 3.2.

Further, the above mentioned work notes that significant improvement to marine Sc (in their models) is not achieved until the vertical grid spacing approaches 15 m in the lower troposphere. Since observational references are not provided in this paper, it's hard to tell if Sapphire's increased vertical grid (which is not as aggressive in the lower troposphere compared to the aforementioned works) significantly improves marine Sc or not. Once observational references are added, it may be interesting to tie the findings with that of the E3SM work.

We already referred to the findings of Bogenschutz et al. (2021) and Lee et al. (2022) in several places of the manuscript, and also mentioned their very small vertical grid spacing of 15 m applied in these studies which is smaller than all our vertical grid spacings. Our comparison to observations shows, however, that the increase of low tropical clouds with higher vertical resolution increases the model bias due to the dominant overestimation of low clouds attributable to the still too coarse horizontal resolution. Hence, we don't add a reference to these papers in the new observation section.

The authors may want to consider citing the work of Cheng et al. (2010) https://doi.org/10.3894/JAMES.2010.2.3 which is a nice paper that lays the groundwork for vertical and horizontal resolution sensitivity for CRMs in the doubly periodic framework.

We agree that referencing this paper is useful, and have done so in the Introduction and Conclusions.

[Figure]

Figure 1: Panels a) to e) show differences of zonally averaged atmospheric quantities between simulations as indicated in the legend of Panel a). The quantities are a) vertically integrated cloud liquid water $\int q_l$, b) vertically integrated cloud ice $\int q_i$, c) total cloud fraction $\int f_{cl}$, d) outgoing shortwave radiation at TOA $F^\uparrow_{SW,TOA}$, and e) outgoing longwave radiation at TOA $F^\uparrow_{LW,TOA}$. Solid lines mark differences of practical vertical resolution changes (i.e. in general also including a time step change), the dashed line a difference caused by a pure vertical resolution change, and the dotted line a difference caused by a pure time step change. All quantities are averaged over the last 40 days of the simulations. Panel f) shows coefficients of correlations between $F^\uparrow_{SW,TOA}$ on the one, and $\int q_l$, $\int q_i$, and $\int f_{cl}$ on the other hand, calculated for each latitude over all experiments of Table 1.

[Figure]

Figure 2: Vertical layer thickness for the examples of grid boxes with surface altitudes of 0 (circles) and 8205 (crosses), respectively. The y-axis shows the height of the lower edge of the respective layer. Colors indicate the experiments (see Table **??** for details) as indicated in the legend. Darker colors mark finer vertical grid spacing. Only the lowest 25 of the grids are shown. The model top is at 75 for all configurations.

[Figure]

Figure 3: Mean vertical profiles of temperature (a) and specific humidity (c) from 257 radio soundings taken between June, 29, and August, 6, 2021 in the tropical Atlantic. Panels b) and d) show differences to these observations from mean simulated profiles from the simulations indicated in the legend and sampled at the time and location of the observations.

[Figure]

Figure 4: Zonally averaged total cloud fraction (a), upward shortwave radiation at the TOA (c), and upward longwave radiation at the TOA (e) from the CERES satellite observations averaged over July, 2021. Panels b), d), and f) show differences of the respective quantities between the simulations indicated in the legend and the satellite products.